# BET inhibitors drive Natural Killer activation in non-small cell lung cancer via BRD4 and SMAD3

Francesca Reggiani [1] ✉, Giovanna Talarico[2,3], Giulia Gobbi[1], Elisabetta Sauta[1,4], Federica Torricelli[1], Veronica Manicardi[1,5], Eleonora Zanetti[6,7], Stefania Orecchioni[2,3], Paolo Falvo [2,3], Simonetta Piana[6,7], Filippo Lococo[8,9], Massimiliano Paci[10], Francesco Bertolini [2,3], Alessia Ciarrocchi [1] & Valentina Sancisi [1] ✉

Non-small-cell lung carcinoma (NSCLC) is the most common lung cancer and one of the pioneer tumors in which immunotherapy has radically changed patients' outcomes. However, several issues are emerging and their implementation is required to optimize immunotherapy-based protocols. In this work, we investigate the ability of the Bromodomain and Extra-Terminal protein inhibitors (BETi) to stimulate a proficient anti-tumor immune response toward NSCLC. By using in vitro, ex-vivo, and in vivo models, we demonstrate that these epigenetic drugs specifically enhance Natural Killer (NK) cell cytotoxicity. BETi down-regulate a large set of NK inhibitory receptors, including several immune checkpoints (ICs), that are direct targets of the transcriptional cooperation between the BET protein BRD4 and the transcription factor SMAD3. Overall, BETi orchestrate an epigenetic reprogramming that leads to increased recognition of tumor cells and the killing ability of NK cells. Our results unveil the opportunity to exploit and repurpose these drugs in combination with immunotherapy.

Non-small-cell lung carcinoma (NSCLC) is the most common type of lung cancer, being divided into two major histological groups: squamous cell carcinoma (SQ) and adenocarcinoma (AD)[1].

Small molecule inhibitors are the standard treatment for NSCLC patients carrying specific mutations, such as *EGFR* or *KRAS* point mutations and *ALK* or *ROS1* rearrangements[2]. Recently, the development of immune-checkpoint (IC) inhibitors has significantly expanded the therapeutic opportunities for lung cancer patients: monoclonal antibodies targeting the PD-1/PD-L1 axis, such as nivolumab and

pembrolizumab, or CTLA4 molecule, such as ipilimumab, are used to treat advanced NSCLC as single agents or in combination with platinum-based chemotherapy[3,4]. However, not all patients seem to benefit from IC inhibition, due to primary or acquired resistance mechanisms[5].

Natural Killer (NK) cells partake in innate immunity and play a key role in anti-viral and anti-tumor responses. NK anti-tumor activity is relevant in NSCLC patients since the extent of NK infiltration positively correlates with overall survival[6–8]. NK cells can exert their cytotoxic

[1]Translational Research Laboratory, Azienda Unità Sanitaria Locale-IRCCS di Reggio Emilia, Reggio Emilia, Italy. [2]Laboratory of Hematology-Oncology, European Institute of Oncology IRCCS, Milan, Italy. [3]Onco-Tech Lab, European Institute of Oncology IRCCS and Politecnico di Milano, Milan, Italy. [4]Humanitas Clinical and Research Center, IRCCS, Rozzano, Milan, Italy. [5]Clinical and Experimental Medicine PhD Program, University of Modena and Reggio Emilia, Modena, Italy. [6]Pathology Unit, Azienda USL-IRCCS di Reggio Emilia, Reggio Emilia, Italy. [7]Biobank, Azienda USL-IRCCS di Reggio Emilia, Reggio Emilia, Italy. [8]Università Cattolica del Sacro Cuore, Rome, Italy. [9]Department of General Thoracic Surgery, Fondazione Policlinico Universitario A. Gemelli IRCCS, Rome, Italy. [10]Thoracic Surgery Unit, Azienda USL-IRCCS di Reggio Emilia, Reggio Emilia, Italy. ✉e-mail: francesca.reggiani2@ausl.re.it; valentina.sancisi@ausl.re.it

function through several mechanisms, comprising the release of perforin and granzymes through exocytic granules or the expression of death-inducing ligands, such as FASL and TRAIL, that triggers apoptosis on target cells. In addition, NK cells produce a set of cytokines, including IFN-γ and TNF-α, which modulates the activity of other tumor microenvironment (TME) immune cells[9,10].

Target cell recognition by NK cells is based on a refined balance of inhibitory and stimulatory signals, mediated by a large set of surface receptors[11]. The inhibitory receptors belong to the HLA-class I-specific inhibitory receptors or Ig-like Receptors (iKIRs) but comprise also NKG2A:CD94, LILRB1, and classical IC receptors, such as PD-1/PD-L1, CTLA4, LAG3, TIM3, and TIGIT[12,13]. NK cytotoxic and anti-tumor activity can be restrained by multiple mechanisms[14,15], including a sustained expression of inhibitory receptors which is detrimental in concomitance with the over-expression of their ligands on cancer cells[16].

Due to their relevant role in targeting tumor cells, adoptive cell therapies with NKs have been proposed for the treatment of both hematologic and solid tumors[17,18]. In addition, NK cells may represent a target for immunotherapy protocols aiming at enhancing/reactivating their anti-tumor properties[13].

Bromodomain and Extra-Terminal protein inhibitors (BETi) are a class of epigenetic drugs that showed promising efficacy in preclinical models of cancers[19,20]. BETi target a bromodomain-containing family of epigenetic reader proteins (BET, including BRD2, BRD3, BRD4, and BRDT), which recognize acetylated histone lysine residues and promote the recruitment of transcription initiation and elongation complexes on target genes[21]. However, preliminary data from clinical trials showed disappointing results with limited efficacy of BETi as single-agent therapy[22]. Nevertheless, evidence of a synergic effect of BETi with immunotherapy was reported in NSCLC models[23]. Mice treated with this drug combination showed robust and long-lasting anti-tumor immune responses compared with either agent alone, and a general improvement in overall survival. In addition, these drugs have been shown to regulate immune response through the inhibition of PD-L1, in several types of cancer, including NSCLC[24,25].

In this work, we investigate the ability of BETi to stimulate a proficient anti-tumor immune response against NSCLC and dissect the molecular mechanism underlying their activity. By using in vitro and ex-vivo models, as well as three different in vivo mouse models, we demonstrate that these epigenetic drugs specifically enhance NK cell cytotoxicity. We also show that BETi down-regulate a large set of inhibitory KIR (iKIR) and IC receptors, that are direct targets of the transcriptional cooperation between BRD4 and SMAD3. Overall, BETi-mediated epigenetic reprogramming leads to increased recognition of tumor cells and consequently increased killing ability of NK cells. Our results unveil the opportunity to exploit and repurpose these drugs in other clinical settings and immunotherapy protocols.

## Results

### BETi enhance NK anti-tumor efficacy against NSCLC cell lines and patient-derived spheroids

The immune regulatory activity of BETi was investigated in patient-derived immune cells isolated from NSCLC surgical specimens (Fig. 1). For the study, we processed samples collected from a total of 53 patients with a new diagnosis of NSCLC, including the two major subtypes AD and SQ.

The immune cell fraction was obtained from the TME after tumor digestion and gradient separation and consisted of a mixed population of tumor-infiltrating lymphocytes, monocytes, and NK cells, that we defined tumor-infiltrating leukocytes (TILs) (Suppl. Fig. 1A, B).

Immune cells were ex vivo treated with BETi epigenetic drugs (1 μM JQ1 or 1 μM OTX015) for 24 h. Cell viability was not affected by the drugs (Suppl. Fig. 1C). TILs were then stimulated with PMA/ionomycin to induce immune cell activation and a general increase of IFN-γ release (Fig. 1A, Suppl. Fig. 1D, E).

BETi induced a significant increase of IFN-γ production in NK cells, compared to other analyzed T lymphocyte populations (Fig. 1B, Suppl. Fig. 1F). This effect was independent on the histopathological subtype of the original patient tumor (AD or SQ).

To challenge the increased anti-tumor activity of BETi-treated immune cells, co-cultures with three different NSCLC cell lines were set up. BETi treatment on immune cells was started 24 h prior the co-culture with tumor cells and maintained throughout the co-culture. The combined administration of BETi with TILs was more efficient in restraining tumor growth (Fig. 1C, Suppl. Fig. 2A, Suppl. Video 1) and enhancing tumor apoptosis (Fig. 1D, Suppl. Fig. 2D, F–H) in all tested cell lines compared to single treatments.

Notably, NK cells purified from the TIL fraction were sufficient to obtain a relevant anti-tumor activity, further supporting the central role of NK cells in triggering anti-tumor response (Fig. 1E, F, Suppl. Fig. 2B, C–E). The increased degranulation activity of NK cells induced by BETi was also confirmed by CD107a surface expression analysis and IFN-γ release in the co-culture medium (Fig. 1G–I, Suppl. Fig. 2I–N), indicating a prominent NK activation toward NSCLC cells.

CD4[+] T lymphocytes purified from the TIL fraction did not show significant anti-tumor activity in these co-cultures, either alone or in the presence of BETi (Suppl. Fig. 3A, D, H, J), whereas purified CD8[+] T lymphocytes displayed relevant cytotoxicity toward NSCLC cells that was enhanced in case of BETi stimulation (Suppl. Fig. 3B, E, I, K). Still, the CD8[+] T cell activation was lower compared to NK activation comparing immune cells isolated from the same patient (Suppl. Fig. 3C, E–G), further supporting the specific effect of BETi in enhancing NK cytotoxicity compared to other TME immune populations.

To further validate the clinical impact of our findings, we isolated primary Cancer Tissue-Originated Spheroids (CTOS), tridimensional models that mimic the original patient's characteristics[26]. We then tested the anti-tumor activity of autologous NK cells through ex-vivo co-cultures in the presence or not of BETi. We chose an intermediate dosage (1 μM) of the drugs that had a limited effect on tumor spheroid integrity (Suppl. Fig. 3L–M). NK cells were red-stained and then co-cultures with CTOS were set up. Tumor 3D growth was monitored through real-time cell imaging (Fig. 2A, Suppl. Video 2). We observed that the NK anti-tumor potential was restored in presence of BETi, with a significant decrease in tumor spheroid growth/integrity in three different AD patients (Fig. 2B, Suppl. Fig. 3N). The complex interactions between receptors/ligands/co-stimulatory molecules taking place at the nanoscale interface between immune and target tumor cells are collectively defined as immunological synapses[27]. The generation of these spatial and molecular interactions between immune and tumor cells is critical to obtain a proficient immune activation. In our co-cultures, we noticed an increased number of immunological synapses between NKs and tumor cells (Fig. 2C) and a concomitant enhanced NK degranulation (Fig. 2D–F).

Collectively, our results indicate that BETi improve the ability of autologous/heterologous NK cells to efficiently recognize tumor cells and enhance their anti-tumor cytotoxic activity.

### BETi treatment transcriptionally represses inhibitory receptors and immune checkpoints (ICs) in NK cells

To get insights into the mechanism through which BETi stimulate NK activation, we took advantage of the NK92 cell line, which is an interleukin-2 (IL-2) dependent line derived from peripheral blood mononuclear cells[28]. NK92 cells display features of activated NK cells and could be used as a valuable tool to study NK biology, showing a similar activation compared to patient-derived NK cells (Suppl. Fig. 4A–C) and toward NSCLC cells (Suppl. Fig. 4D–K, Suppl. Video 3 and Suppl. Video 4).

To dissect if BETi-mediated increased cytotoxicity was dependent on specific regulatory circuits, we performed RNA sequencing (RNA-seq) comparing vehicle- and BETi-treated (1 μM OTX015) NK92 cells.

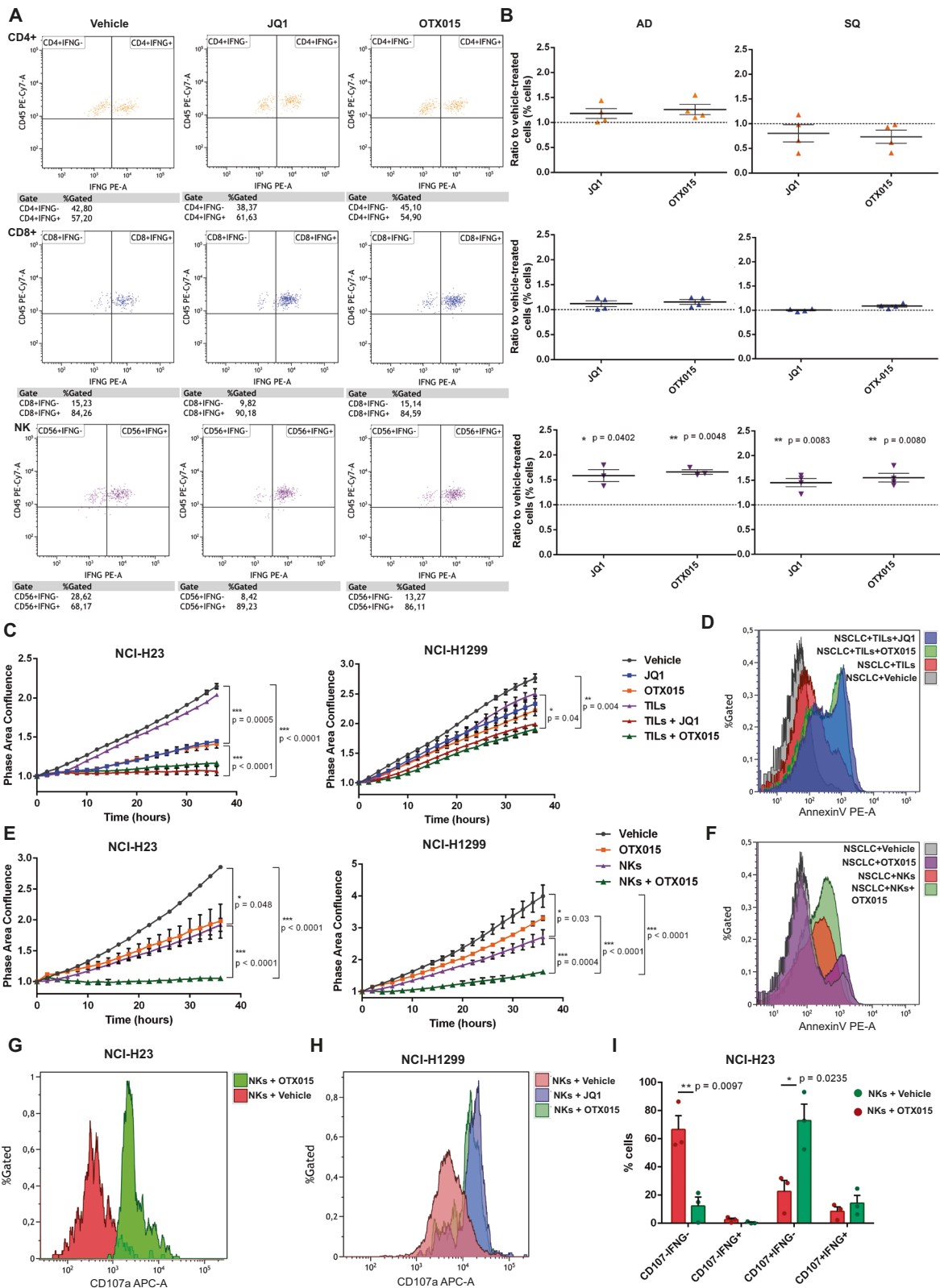

1484 genes were detected as significantly up-regulated after BETi treatment, whereas 1553 genes were down-regulated (Fig. 3A, Suppl. Fig. 5A). Gene ontology (GO) and Reactome enrichment analysis identified several immune-related and inflammatory pathways that were up- or down-regulated by BETi (Fig. 3B, C, Suppl. Fig. 5B, C). STRING network analysis of top-scoring down-regulated genes indicated that they have a prevalent membrane surface localization and

are involved in cell-to-cell communication (Fig. 3B). Conversely, top-scoring up-regulated genes encode molecules with membrane-bound organelle localization and a major role in cell metabolism or chromatin remodeling (Suppl. Fig. 5C).

Among the top enriched down-regulated pathways, we detected cytokine-mediated signaling (R-HSA-1280215; GO:0019221; GO:0071345; GO:0034097) and immune system regulatory pathways

**Fig. 1 | BETi stimulate NK cytotoxicity toward NSCLC in vitro. A**, **B** Flow cytometry analysis of different immune cells isolated from NSCLC patient surgical samples after PMA/Ionomycin stimulation for 6h. Gating of IFN-γ⁺ helper (CD4⁺), cytotoxic (C8⁺) T cells or NK cells (CD56⁺) were analysed for IFN-γ production (**A**). Quantitation of IFN-γ⁺ immune cells (**B**) indicates that NK cells displayed a higher increase of IFN-γ⁺ cells when treated with BETi compared to vehicle, irrespectively of the tumor histopathological features from which they are isolated (AD $n = 4$ patients, SQ $n = 4$ patients, data are shown as mean ± SEM); **C** Proliferation assays on different NSCLC cell lines (NCI-H23, NCI-H1299) in co-culture with patient-derived total CD45⁺ cells (Tumor-infiltrating leukocytes, TILs) and BETi (1 μM JQ1 or OTX015). Tumor cell confluence area was measured by EssenBio Incucyte S3 cell-live imaging. BETi significantly improved immune response, by restraining tumor cell proliferation compared to single-agent treatments ($n = 3$ independent experiments, data are shown as mean ± SEM); **D** Apoptotic tumor cells (AnnexinV⁺) were assessed by flow cytometry after 24 h of co-cultures with TILs in presence or not of

BETi. Tumor apoptosis (NCI-H1299) was higher in BETi-treated co-cultures. **E** Purified patient-derived NK cells were sufficient to inhibit tumor proliferation in co-cultures with NSCLC and their effect was enhanced by BETi ($n = 6$ biological independent samples, data are shown as mean ± SEM); **F** Tumor apoptosis enhancement by flow cytometry in co-cultures between purified NK cells and NCI-H1299 cells; **G**–**I** Flow cytometry analysis of NK cell activation markers from co-cultures with NSCLC cells (NCI-H23, NCI-H1299). Surface expression of CD107A (degranulation marker) in NK cells was assessed 24 h after the starting of the co-cultures and was enhanced in presence of BETi (**G**, **H**). Combining the quantitation of IFN-γ and CD107A, BETi specifically enhanced the percentage of CD107A⁺IFN-γ⁻ NK cells, whereas the increase of double positive IFN-γ⁺ CD107A⁺ was not significant in co-cultures with NCI-H23 ($n = 3$ patients, data are shown as mean ± SEM) (**I**). Two-side Student's $t$-test was applied for all comparisons in this figure. Source data are provided as a Source Data file.

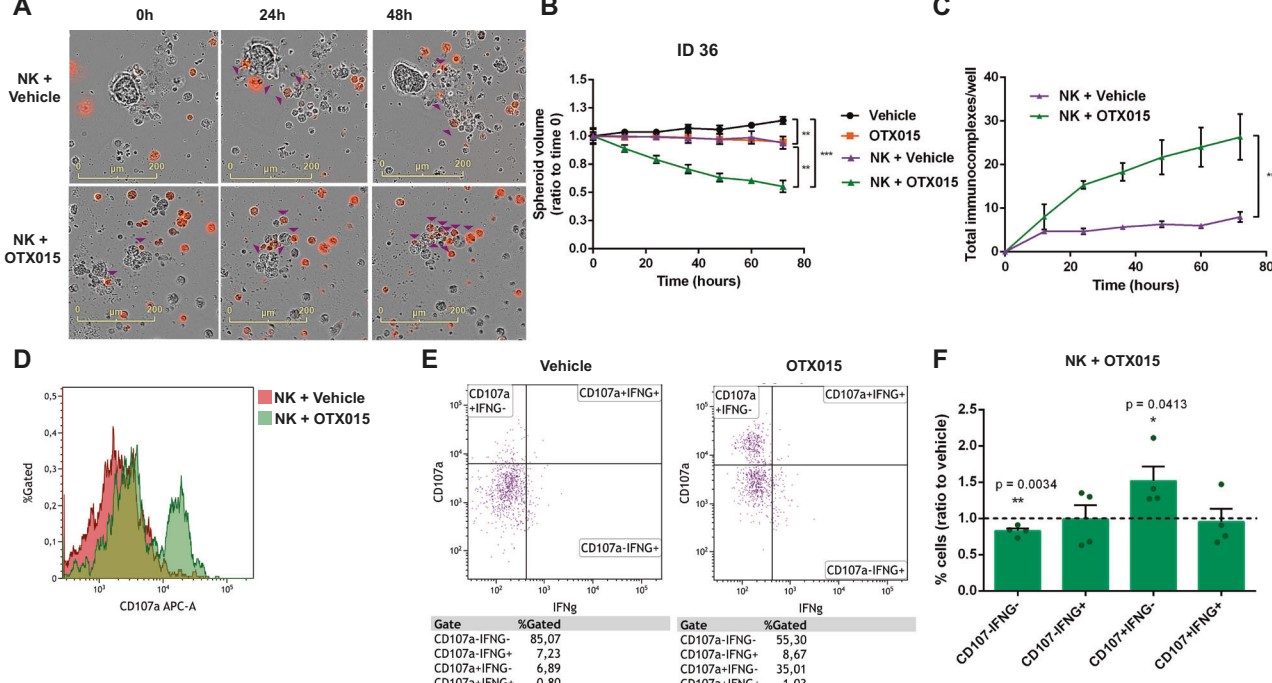

**Fig. 2 | BETi reactivate autologous NK cytotoxicity toward patient-derived spheroids. A** Representative images of CTOS derived from NSCLC surgical specimens used in co-cultures with autologous red-stained NK cells (500 NK cells/well) acquired with Incucyte 2020B software (10x magnification, scale bar 200 μm). Purple arrows indicate NK cells that are forming immunological synapses with tumor cells; **B** Quantitation of CTOS 3D-growth in these co-cultures. Spheroid volume was calculated at each time point and normalized for T0 volume. Autologous NK cells were pre-treated with BETi (1 μM OTX015) or vehicle 24 h before the co-culture and during the co-culture. BETi-treated NK cells were significantly more efficient in dissolving CTOS integrity compared to untreated NK cells or BETi alone ($n = 6$ spheroids per group from a single AD patient, error bars indicate mean ± SEM. Vehicle vs. OTX015 ***$p = 0.0008$; Vehicle vs. NKs + Vehicle **$p = 0.0012$;

Vehicle vs. NKs + OTX015 ***$p = 0.0003$; OTX015 vs. NKs + OTX015 **$p = 0.0022$; NKs + Vehicle vs. NKs + OTX015 **$p = 0.0021$); **C** Quantitation of immunological synapse formation in these autologous 3D co-cultures ($n = 3$ co-cultures per group from a single AD patient, data are shown as mean ± SEM. NKs + Vehicle vs. NKs + OTX015 **$p = 0.0098$); **D**–**F** Flow cytometry analysis of NK cell activation after 24 h of co-cultures with autologous CTOS. Surface expression of CD107A showed that BETi enhanced NK degranulation as increased CD107A Mean Fluorescence Intensity (MFI) (**D**) or percentage of CD107A⁺ cells (**E**). Combining the quantitation of IFN-γ and CD107A (**F**), BETi enhanced the percentage of CD107A⁺IFN-γ⁻ NK cells, whereas double negative IFN-γ⁻ CD107a⁻ cells were impaired ($n = 4$ AD patients, data are shown as mean ± SEM). Two-side Student's $t$-test was applied for all comparisons in this figure. Source data are provided as a Source Data file.

(R-HSA-168256; R-HSA-1280218; GO:0050868), which include a large set of IC molecules and iKIR receptors (Reactome enrichment Fig. 3C, GO enrichment Suppl. Fig. 5B). IC receptors whose expression was inhibited by BETi include classical PD-1/PD-L1 and CTLA4 axes, as well as the second-generation IC molecules, such as LAG3, TIGIT, CD80, CD83, CD86, HAVRC2, and CEACAM1. iKIR molecules that are known to play an important role in NK self-tolerance (i.e. KIR2DL1, KIR3DL3, KIR2DL3, etc.) were also significantly down-regulated by BETi.

The reduced expression of ICs and iKIRs after BETi treatment was confirmed in an independent set of NK92 samples (Fig. 3D) NSCLC-

purified primary NK cells (Fig. 3E). In addition, a significantly reduced expression of ICs was confirmed by flow cytometry analysis in both patient-derived intratumor NKs and NK92 cells (Suppl. Fig. 5D–F). The dose-response analysis to JQ1 or OTX015 drugs in NK92 cells indicated that the BRD4 targets, including the previously known targets such as c-MYC and COL1A1 and the targets identified in this manuscript (TIGIT, LAG3), were significantly down-regulated in a dose-dependent manner by BETi starting from a 0.5-1 μM concentration after 24 h (Suppl. Fig. 5G–H).

To confirm that this transcriptional regulation was a direct effect of the inhibition of the BET protein BRD4 and not just the consequence

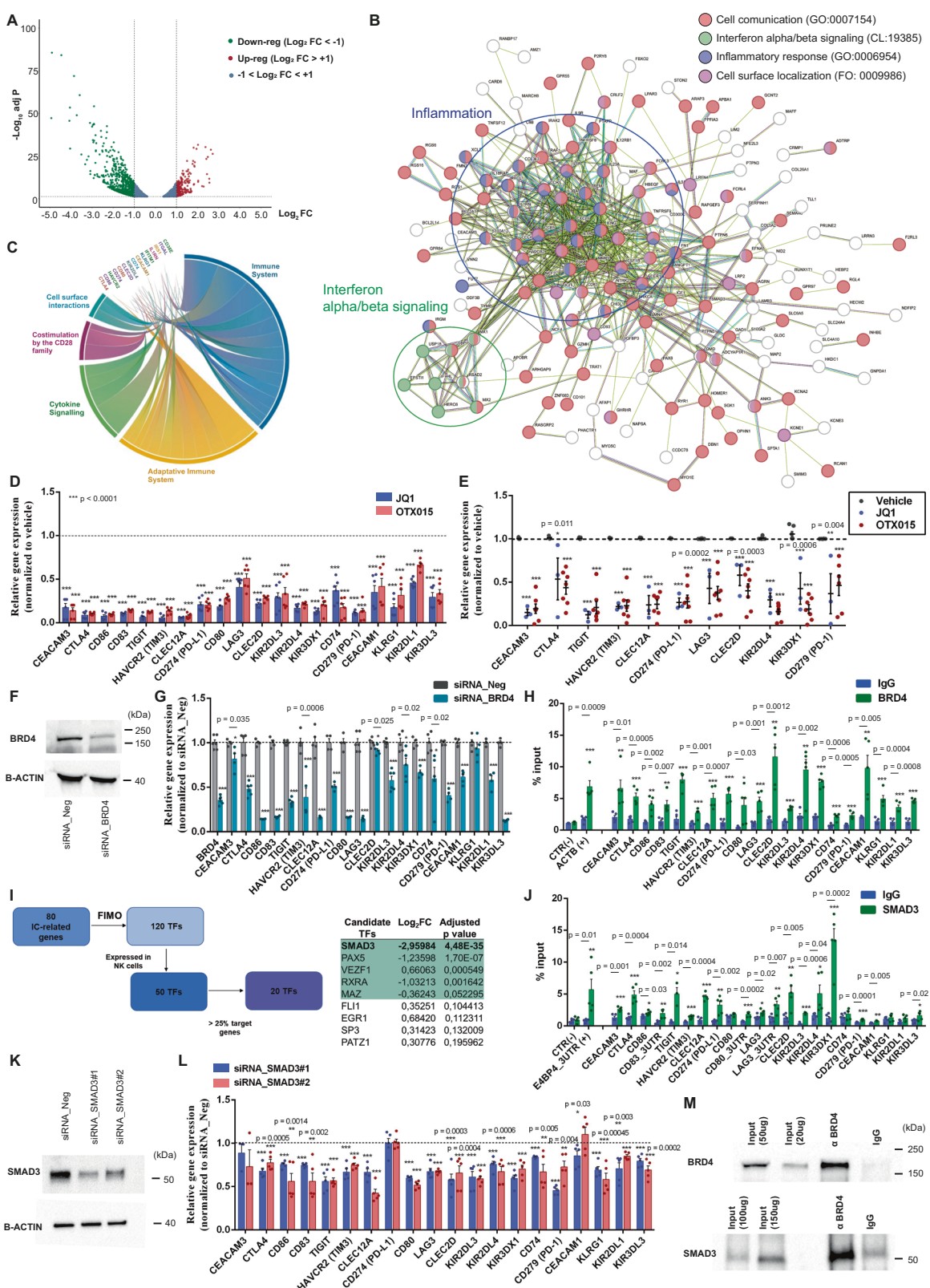

of a genome-wide transcriptional reorganization induced by the epigenetic drugs, we obtained NK92 carrying the Knock-Down (KD) of BRD4 (BRD4$^{KD}$) (Fig. 3F). BRD4 silencing was sufficient to downregulate the expression of iKIR and IC molecules in NK cells (Fig. 3G). To verify the direct transcriptional regulation of BRD4 on these targets, we performed Chromatin Immunoprecipitation (ChIP) analysis

that confirmed BRD4 enrichment on their Transcription Starting Site (TSS) in unstimulated cells (Fig. 3H). To further corroborate these results, we analyzed publicly available ChIP-seq datasets for BRD4 in peripheral-blood NKs (GSE156423). We confirmed BRD4 enrichment within 2 kb window around the center of our ChIP amplified regions (Suppl. Fig. 6A). We observed that among these BRD4 targets, only a

**Fig. 3 | BETi orchestrate the down-regulation of an NK exhaustion signature which is dependent on SMAD3. A** Volcano plot displaying significantly DEGs between OTX015- and vehicle-treated NK92 (3037 genes with adjusted *p*-value < 0.05; *Green* = Down-regulated genes with Log$_2$FC < −1, *Red* = Up-regulated with Log$_2$FC > 1, *Blue* = genes with |Log$_2$FC| < 1); **B** STRING Network analysis of top-scoring down-regulated genes from RNA-seq (Log$_2$FC ≤ − 1.0). Genes belonging to enriched GO pathways are highlighted; **C** Chord diagram illustrating top-scoring down-regulated genes belonging to immune pathways from enrichment analysis using Reactome (Immune system R-HSA-168256; Adaptive Immune system R-HSA-1280218; Cytokine Signaling R-HSA-1280215; Costimulation by CD28 family R-HSA-388841; Cell surface interactions R-HSA-202733). The arch size is proportional to gene deregulation; **D** IC/iKIR expression in BETi-treated NK92 by qPCR. Values were normalized for housekeeping gene expression and expressed as fold change (*n* = 6 biological independent samples); **E** The reduced expression of a selection of ICs/iKIRs was validated by qPCR in purified NKs from surgical samples (*n* = 7 patients); **F** Representative WB depicting BRD4$^{KD}$ (-200 kDa) 72 h after transfection with

siRNAs. *β*-actin was used as loading control; **G** IC expression in NK92 carrying BRD4$^{KD}$ by qPCR (*n* = 6 biological independent samples); **H** ChIP of BRD4 identified its binding on IC/iKIR promoter regions in NK92. Actin B (*ACTB*) promoter was used as positive control, whereas an intergenic region was the negative control (CTR-). Values are expressed as % of input (*n* = 5 biological independent samples); **I** Prediction of candidate transcription factors (TFs) that regulate the 80-gene signature associated with NK exhaustion. TF ranking was obtained according to the adjusted *p*-value and log$_2$FC from RNA-seq in NK92; **J** ChIP of SMAD3 on signature gene promoters or 3'UTRs in NK92. *E4BP4* 3'UTR was used as positive control (*n* = 5 biological independent samples); **K** Representative WB of SMAD3 (-52 kDa) in NK92 after siRNA-mediated SMAD3$^{KD}$; **L** IC/iKIR expression in SMAD3$^{KD}$ NK92 by qPCR (*n* = 5 biological independent samples); **M** Representative images of co-immunoprecipitation (Co-IP) of SMAD3 and BRD4 in NK92. All data are shown as mean ± SEM. Two-side Student's *t*-test was applied for all comparisons in this figure. If not specified, ***p ≤ 0.0001. Source data are provided as a Source Data file.

subset was positive for BRD4 binding in K562 cells (GSE101225)[29], suggesting a specific transcriptional program exerted by BRD4 in NK cells (Suppl. Fig. 6A).

## BRD4 cooperates with SMAD3 in controlling the NK exhaustion signature

BRD4 is an epigenetic reader that regulates the transcription of target genes by fostering the recruitment of other transcription factors (TFs) on specific regulatory elements. Among DEGs from our RNA-seq, we identified 80 genes encoding for iKIRs, classical and non-classical ICs, and IC-related molecules. All of them were down-regulated by BETi and are associated with an NK exhaustion profile, providing a specific gene signature. By applying the FIMO algorithm (Fig. 3I), we scanned the promoter regions of these genes searching for TFs binding motifs. We predicted a total of 120 putative TFs whose binding motif was enriched and that were candidate upstream regulators of the NK exhaustion signature. Of these, 50 TFs were confirmed to be expressed in NK cells, based on our RNA-sequencing results. TFs that recognized <20 genes (i.e. <25% of NK exhaustion signature) were excluded, generating a list of 20 top-scoring candidates. These TFs were further prioritized based on adjusted *p*-value and the differential expression in our RNA-seq analysis. SMAD3 emerged as the top-scoring candidate. To validate the FIMO prediction, we performed ChIP-target analysis for SMAD3 on the promoter regions of the signature genes that we found enriched by BRD4 binding (Fig. 3J). We confirmed SMAD3 binding enrichment to the promoters of most of these genes, suggesting that SMAD3 has a direct function in their transcriptional regulation. Of interest, we detected SMAD3 binding on the 3'UTR regions of some genes belonging to the signature, suggesting a more complex layer of regulation (Fig. 3J). Coherently, NK92 cells carrying SMAD3$^{KD}$ (Fig. 3K) displayed the down-regulation of many genes from our signature, confirming a concerted regulation with BRD4 (Fig. 3L). Since we demonstrated that SMAD3 and BRD4 co-occupied the same genomic regions, we further demonstrated that SMAD3 directly interacts with BRD4 in NK92 nuclear fraction by co-immunoprecipitation (co-IP) analysis (Fig. 3M).

To further corroborate the transcriptional cooperation between BRD4 and SMAD3, we compared ChIP-seq public datasets for both BRD4 and SMAD3 in K562 cells from the ENCODE project (GSE231137 for SMAD3, GSE101225 for BRD4)[30]. Intriguingly, -25% of BRD4 peaks overlapped SMAD3 peaks (Suppl. Fig. 6C). Moreover, the meta-profile analysis revealed a relevant enrichment of BRD4 signal on SMAD3 peak regions (Suppl. Fig. 6D). In line with these results, a positive correlation between BRD4 and SMAD3 enrichment at the TSS of protein-coding genes was observed (Suppl. Fig. 6E).

Collectively, these data indicate that SMAD3 and BRD4 cooperate to support the expression of an NK-specific inhibitory signature.

## BRD4 exerts a robust transcriptional control on SMAD3 expression in NK cells

According to the differential expression of SMAD3 observed in our RNA-seq, we confirmed that SMAD3 expression was significantly down-regulated by BETi treatment (1 μM JQ1 or 1 μM OTX015) in both NK92 and patient-derived NK cells (Fig. 4A–C). Notably, this transcriptional inhibition was an early event occurring already 3 h after BETi treatment in NK92 cells (Fig. 4D). The significant down-regulation of SMAD3 expression was also observed at lower doses of BETi, starting from 50 nM at 24 h (Suppl. Fig. 5G, H). These evidences suggest that SMAD3 can be not just a cooperator, but also a direct target of BRD4 transcriptional regulation. We confirmed this hypothesis by identifying multiple BRD4 binding sites on *SMAD3* promoter regions by ChIP analysis (Fig. 4E, F, Suppl. Fig. 6F). We also showed that BETi treatment deeply affected BRD4 recruitment in *SMAD3* genomic regions by detaching its binding (Fig. 4G, Suppl. Fig. 6G). Indeed, the BRD4$^{KD}$ in NK92 cells leads to significant repression of SMAD3 expression, recapitulating the effect of BETi treatment (Fig. 4H). These results were further corroborated by the comparison with ChIP-seq publicly available datasets for BRD4 in which *SMAD3* promoter regions were detected among top-scoring BRD4 targets, in both NKs and K562 cells (Suppl. Fig. 6A, B, Suppl. Fig. 6H).

Collectively, we speculate that BRD4 controls SMAD3 at multiple levels. SMAD3 expression was regulated by BRD4 through direct transcriptional regulation, whereas SMAD3 transcriptional activity was driven by BRD4 interaction on chromatin target sites. This multi-layer functional cooperation between BRD4 and SMAD3 was further supported by ChIP analysis on NK92 cells treated with BETi. The epigenetic drugs detached both BRD4 (Fig. 4I, Suppl. Fig. 6I) and SMAD3 (Fig. 4J, Suppl. Fig. 6J) from their genomic targets, restraining their transcriptional activity.

## SMAD3/BRD4 inhibition promotes target cell recognition and NK activation toward NSCLC

To assess the role of SMAD3 in controlling NK cytotoxicity and activation, we performed 3D co-culture experiments with SMAD3$^{KD}$ NK92 and NSCLC cell line-derived spheroids.

The overall ability of SMAD3$^{KD}$ NK92 to restrain tumor spheroid growth was enhanced compared to control cells (Fig. 5A, Suppl. Fig. 7A), but lower as compared to BETi-treated or BRD4$^{KD}$ carrying NK92 cells (Fig. 5B, Suppl. Fig. 7B). This indicates that BETi and BRD4 modulated additional NK regulatory pathways besides the ones that were dependent on SMAD3.

Notably, SMAD3$^{KD}$ was sufficient to increase the immunological synapse formation toward NSCLC in the early phases of co-cultures, closely mimicking the observed effect of BETi-treated (1 μM OTX015) or BRD4$^{KD}$ NK92 cells (Fig. 5C–F, Suppl. Fig. 7C–F). These results were further validated using purified intra-tumor NKs from AD patients in

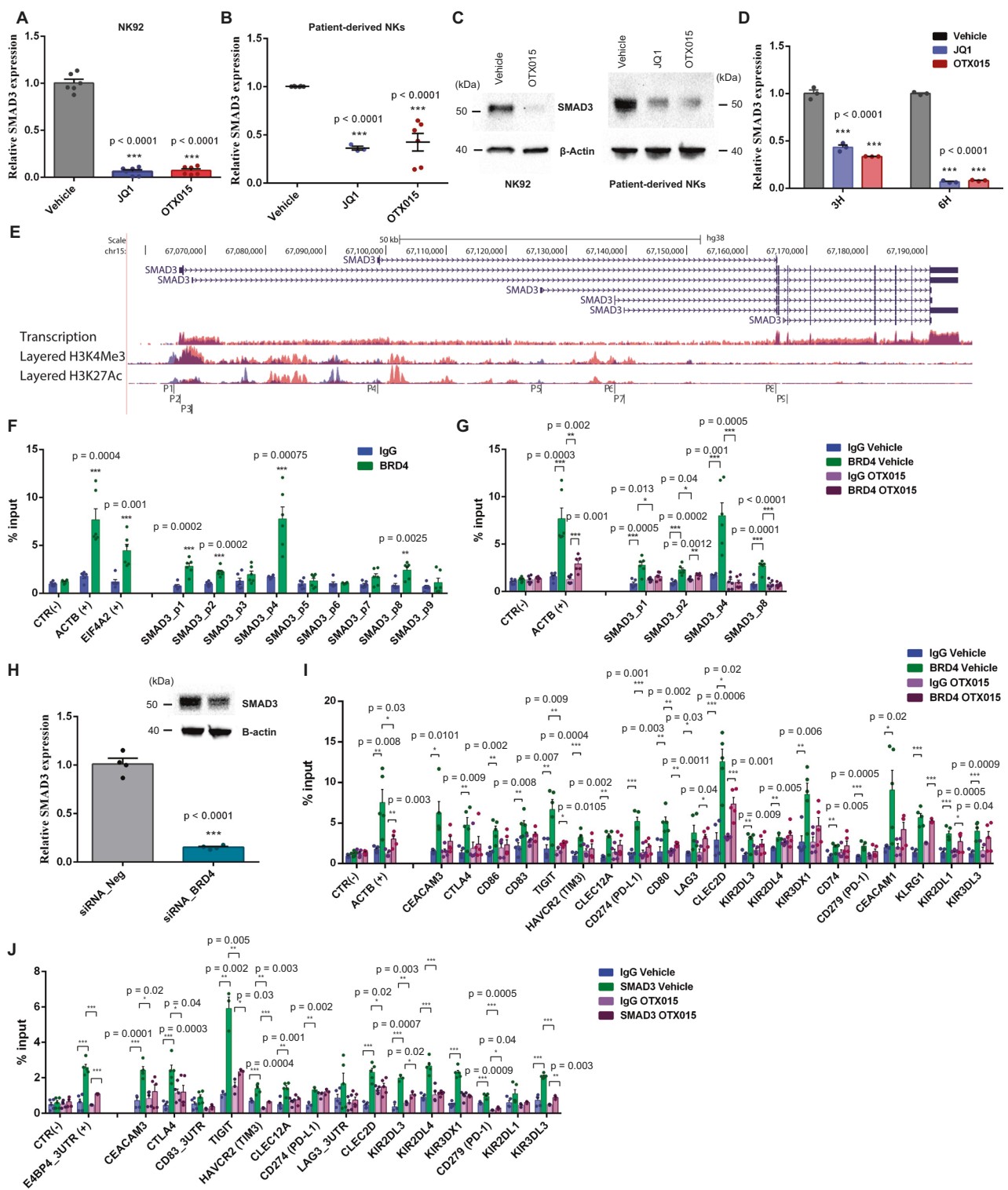

which the SMAD3[KD] cells displayed a reduced expression of the main ICs and KIRs, an increased ability to contrast NSCLC cells and an improved formation of immunological synapses compared to control cells (Fig. 5G–I, Suppl. Fig. 7G).

To further decipher the role of SMAD3 in mediating BRD4 activity, we assessed the impact of its upstream inducer Transforming Growth Factor β (TGF-β) in controlling NK effector properties. We confirmed that the TGF-β known transcriptional targets[31,32] were reduced after 6 h of stimulation (Suppl. Fig. 7H). We also observed that administration of TGF-β induced a rapid nuclear translocation of

SMAD3 in NK92 cells, promoting its transcriptional activity (Suppl. Fig. 7I). TGF-β stimulation was sufficient to induce IC and iKIR expression in NKs and triggered the rescue of the BETI-mediated down-regulation of these BRD4/SMAD3 common targets (Fig. 5J). Accordingly, the pre-treatment of NK92 cells with TGF-β impaired the formation of immunological synapses in the early phase of our co-cultures, restraining the observed BETi ability to improve target tumor cell recognition and further supporting the central role of SMAD3 inhibition in mediating BETi anti-tumor effects (Fig. 5K, Suppl. Fig. 7J).

**Fig. 4 | BRD4 regulates SMAD3 expression in NKs by direct transcriptional control. A**, **B** Reduction of SMAD3 expression in NK92 (**A**) or patient-purified NKs (**B**) after 24 h of BETi treatment detected by qPCR. Values are normalized for housekeeping gene expression and expressed as fold change (**A** $n = 6$ biological independent samples, **B** $n = 6$ patients, data are shown as mean ± SEM); **C** Representative WB of SMAD3 (52kDa) in NK92 or patient-purified NKs after BETi. $\beta$-actin was the loading control; **D** Time course of SMAD3 reduced expression assessed by qPCR in NK92 treated with BETi ($n = 3$ independent experiments, data are shown as mean ± SEM); **E** Illustration of the genomic region encoding for SMAD3. The layered transcription, H3K4Me3 and H3K27Ac tracks are relative to GM12878 (*orange*) and K562 (*violet*) cells. The highlighted promoter sequences (P1-P9) were analysed in NK92 by ChIP to detect BRD4 binding. Images were modified from Genome Browser; **F** ChIP identified BRD4 binding on promoter regions of *SMAD3* in NK92. *ACTB* promoter was the positive control, whereas an intergenic region was used as negative control (CTR⁻). Values are expressed as % of input ($n = 6$ biological independent samples, data are shown as mean ± SEM); **G** Treatment with 1 μM OTX015 for 48 h induced BRD4 detachment from *SMAD3* promoter regions ($n = 6$ biological independent samples, data are shown as mean ± SEM); **H** SMAD3 transcription was inhibited in BRD4$^{KD}$ NK92 as detected by qPCR ($n = 4$ independent experiments, data are shown as mean ± SEM) and by WB; **I** ChIP analysis of BRD4 in NK92 cells treated with OTX015 for 48 h identified its detaching from target regulatory regions ($n = 5$ biological independent samples, data are shown as mean ± SEM); **J** ChIP analysis of SMAD3 in NK92 treated with OTX015 for 48 h indicated the reduction of the TF occupancy on NK exhaustion gene promoters or 3'UTRs. *E4BP4* 3'UTR was the positive control ($n = 5$ biological independent samples, data are shown as mean ± SEM). Two-side Student's $t$-test was applied for all comparisons in this figure. If not specified, ***$p \leq 0.0001$. Source data are provided as a Source Data file.

Altogether, these results further supported the role of SMAD3 in controlling tumor recognition by NK cells through active cooperation with the epigenetic reader BRD4 in regulating inhibitory receptors and NK exhaustion genes.

## BETi down-regulated distinct immune-checkpoint axes targeting both NKs and tumor cells

To further assess the impact of IC molecule inhibition promoted by BETi in the context of NK cytotoxicity, we compared the efficacy of these epigenetic drugs with monoclonal antibodies targeting different IC receptors, including PD-1 (Nivolumab), CTLA4 (Ipilimumab) or TIGIT (Vibostolimab) in our co-culture system with NSCLC cell lines and patient-derived intratumor NKs or NK92 cells (Fig. 5L, Suppl. Fig. 8). OTX015 alone displayed the highest efficacy in stimulating the NK anti-tumor activity compared to single-agent IC inhibitors. Moreover, the combination of IC inhibitors with BETi did not further enhance NK cytotoxicity toward NSCLC cells. This is in line with our observations that BETi induced a multiple IC down-regulation in NKs.

Of note, the BETi-mediated down-regulation of ICs observed in NKs was associated with a concomitant reduction of respective IC ligands in tumor cells (such as PD-L1, PD-L2, LSGAL9, HVEM), both NSCLC cell lines and patient-derived tumors (Suppl. Fig. 9A–E). Moreover, we observed that BETi triggered the down-regulation of HLA class I molecule expression, especially HLA-F, thus further increasing the overall ability of NKs to recognize tumor target cells (Suppl. Fig. 9A–E). PD-L1 inhibition in NSCLC cell lines was achieved at a micromolar dosage, using both JQ1 or OTX015, and was in line with the induced inhibition of other BRD4 targets, such as c-MYC (only for H1299 cells), RUNX2, COL1A1 and RAD51 (Suppl. Fig. 9F–I).

Collectively, our data indicated that BRD4 plays a relevant role in coordinating a complex network of transcriptional regulation impacting immune response, by acting both on NKs and tumor cells, controlling IC signaling and self-molecule expression and negatively affecting the anti-tumor activity of NKs.

## The anti-tumor efficacy of adoptive NK cell therapy is enhanced by BETi in vivo

To increase the clinical impact of our findings, we tested the combination of BETi with an NK-based adoptive cell therapy using two xenograft models of NSCLC (Fig. 6A).

NCI-H23-LUC cells were subcutaneously injected in NSG mice and tumor growth was monitored by digital caliper and bioluminescence-based imaging (Fig. 6B–E). When tumors became detectable by luminescent analysis by IVIS system (~2 weeks after tumor injection), the intravenous administration of NK92 cells alone or in combination with intraperitoneal administration of OTX015 was applied. The combined treatment displayed the most significant effect in reducing tumor growth, compared to vehicle or single-agent therapies.

A second orthotopic model was generated with NCI-H1299-LUC cells, which were intravenously administered and grafted to lungs in 1 week. When a basal luminescent signal was observed in the lungs, single or combined therapies were started. According to our previous model, the efficacy of the adoptive administration of NK92 cells was maximized in presence of BETi, restraining tumor growth and local invasion (Fig. 6D–G). Tumor lung lesions were analysed by pathological assessment (Fig. 6H–J). NCI-H1299-LUC generated multicentric and multifocal lung lesions localized near blood vessels. Both the number of tumor foci and the major axis of single lesions were significantly reduced in single- and combined-treated mice compared to vehicle controls. The combined treatment seems to be particularly efficient in limiting tumor lesion extension and invasion into the surrounding lung tissue. No metastases were observed in all mice.

To exclude that the observed increased anti-tumor efficacy was caused by differential NK recruitment in BETi-treated mice, we investigated the presence of NK92 cells by flow cytometry (Fig. 6K, L) and IHC analysis (Fig. 6M, N). BETi did not affect the number of NK92 cells in both mouse models.

Overall, these results further support the use of BETi to increase the anti-tumor efficacy of NK cells in vivo and strengthen the rationale of their combination with immunotherapy-based protocols.

## NK cells are required to obtain an efficient BETi anti-tumor response in vivo syngeneic models

To fully demonstrate that NKs are essential mediators of BETi immune regulation in NSCLC, we investigated BETi effects in Lewis Lung carcinoma (LLC1) syngeneic model obtained in the C57-BL/6 background (Fig. 7A). LLC1-LUC cells were subcutaneously injected and tumor growth was monitored by bioluminescence-based imaging. When tumors became detectable by luminescent analysis (~1 week after tumor injection), OTX015 was administered to mice with or without an NK-neutralizing antibody (α mouse CD122) that induced NK cell depletion.

The results indicated that NKs were relevant immune players in the NSCLC anti-tumor immune surveillance and their depletion was sufficient to boost tumor growth in these models (Fig. 7B–E). Of note, NK depletion completely neutralized the OTX015-mediated anti-tumor response, suggesting that NK presence was fundamental in predicting BETi efficacy (Fig. 7B–F).

We did not observe a perturbation in the percentage of different immune cell subsets in mouse tumors comparing OTX015- to vehicle-treated mice, with only a trend of increase of tumor-infiltrated NKs that did not reach statistical significance (Fig. 7G, H). Conversely, the activation induced by OTX015 appeared highly specific for the NK subset compared to the other cytotoxic CD8⁺ T population, as demonstrated by the increased degranulation marker expression on NKs (Fig. 7I, J). According to our in vitro/ex vivo data, a concomitant reduction of IC expression of PD-1, CTLA4, and TIGIT was detected on NKs from OTX015-treated mice, confirming the reduction of NK exhaustion profile (Fig. 7K, L).

Collectively, these data confirmed that NKs are fundamental to support the anti-tumor activity of BET inhibitors in vivo and that BETi

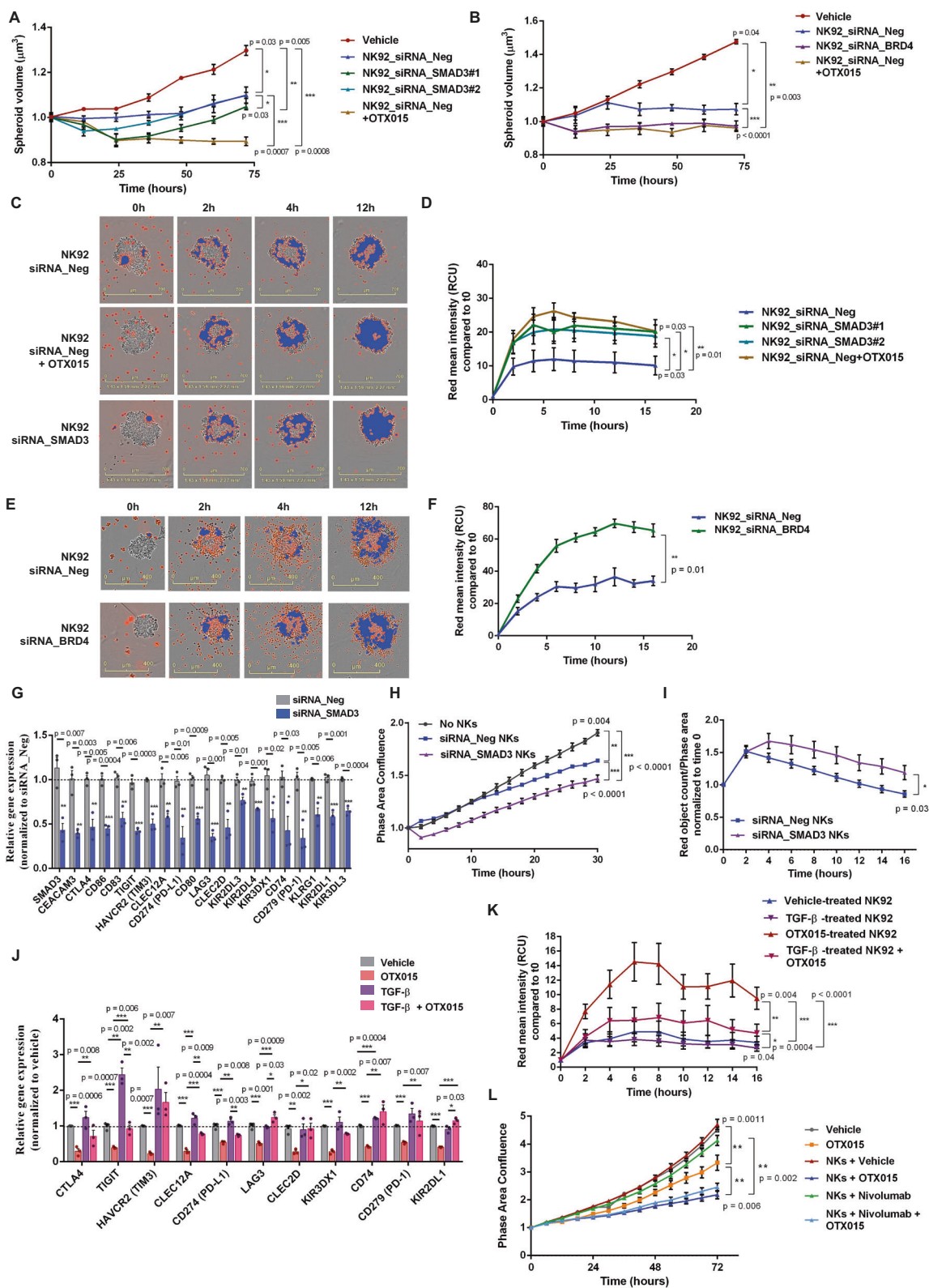

immune regulation was strictly dependent on the presence of activated and functional NKs.

## Discussion

Immunotherapy has radically changed the therapeutic approach and perspective for NSCLC patients[4,5]. However, several issues are emerging as critical steps toward the full implementation of immune-based protocols, based on either IC blockade or adoptive immune cell therapy administration. Among the major limitations to their application, the acquired resistance and the lack of long-lasting complete response due to improper stimulation of immune cells remain urgent unmet clinical needs[33]. The study of TME and immune populations is thus intimately connected with the identification of therapeutic targets for immunotherapy-based approaches and the

**Fig. 5 | SMAD3^KD or BRD4^KD increases NK immunological synapses toward NSCLC.** **A** 3D-growth assays on NCI-H23 spheroids in co-culture with red-stained SMAD3^KD NK92. Spheroid volume was calculated at each time point. NKs were pre-treated with OTX015 or vehicle 24 h before and during co-culture. SMAD3^KD increased NK92 anti-tumor efficiency but to a less extent than OTX015 (n = 6 spheroids per group); **B** 3D-growth assays on NCI-H23 spheroids in co-culture with red-stained BRD4^KD NK92 (n = 6 spheroids per group); **C, D** Immunological synapse formation between red-stained NK92 and NCI-H23 spheroids was assessed by quantifying Red Fluorescence mean intensity (RCU) within the brightfield area of each spheroid (4 x magnification, scale bar 600 μm). Blue masks correspond to quantified areas. SMAD3^KD increases NK92 ability to form immunological synapses, similarly to BETi (n = 4 spheroids per group); **E, F** Immunological synapse formation between red-stained BRD4^KD NK92 and NCI-H23 spheroids (4x magnification, scale bar 400 μm). BRD4^KD enhanced immunological synapse formation (n = 7 spheroids per group); **G** SMAD3^KD obtained by siRNAs in patient-derived intratumor NKs down-regulated the expression of ICs/iKIRs (n = 3 patients); **H, I** Representative proliferation assay of co-cultures using SMAD3^KD primary NKs and NCI-H23 cells (**H**, n = 4 co-cultures from a single patient; **I** n = 5 co-cultures from a single patient). NKs displayed an increased anti-tumor activity (**H**) and an improved ability to form immunological synapses (**I**); **J** qPCR analysis of SMAD3/BRD4 common target expression in NK92 after 6 h treatment with the SMAD3 inducer TGF-β (5ng/ml) or OTX015 (1 μM) or their combination (n = 3 independent experiments); **K** Pre-treatment with TGF-β (5 ng/ml) or OTX015 (1 μM) or both was administered to NK92 16 h prior the starting of the co-culture with NCI-H23. TGF-β impaired NK ability to recognize tumor cells and immunological synapse formation in the early phases of the co-cultures, drastically affecting OTX015 efficacy (n = 8 spheroids per group); **L** Representative proliferation assay of co-cultures using patient-derived intratumor NKs and NCI-H23 cells. NK anti-tumor activity was assessed in the presence of BETi (1 μM OTX015) and/or the anti-PD-1 inhibitor Nivolumab (15 μg/ml) (n = 3 co-cultures from a single patient). All data are shown as mean ± SEM. Two-side Student's t-test was applied for all comparisons in this figure. If not specified, ***p ≤ 0.0001. Source data are provided as Source Data file.

combination with other treatments to establish proficient anti-tumor immunity.

In this context, our work provides an original perspective on the clinical application of the epigenetic drugs BETi to NSCLC patients. Here, we demonstrated that BETi drive a potent anti-tumor immune response able to trigger NK cell cytotoxicity.

BETi target BET proteins, a family of epigenetic readers, among which BRD4 is the most studied member. BRD4 binds acetylated histones of active promoters and enhancers and drives oncogenic transcription programs in several neoplastic diseases[34–37]. For these reasons, BETi were proposed as anti-cancer agents. However, they showed limited efficacy as single-agent therapy, hampering their clinical applicability[22]. Notably, the most efficient therapeutic effect of BETi was achieved in mouse models with an intact host immune system, suggesting a role in immune regulation[25].

In this work, we first investigated BETi ability to stimulate immune cells purified from the TME of NSCLC patients, including both adenocarcinoma and squamous cell carcinoma subtypes. We demonstrated that BETi selectively induce the activation and anti-tumor potential of NK cells toward tumor cells.

This is in line with other studies which reported BETi ability to modulate TME inflammation, immune surveillance and NK cytotoxicity[38,39]. NK cells were required in small-cell lung cancer (SCLC) to mediate BETi anti-tumor response[40] and in NSCLC models to maximize IC blockade and BETi synergistic activity[23].

As previously reported in other studies[41–43], we observed that BETi enhanced the cytotoxicity of intratumor CD8+ T-Lymphocytes. However, we also noted that BETi induction of immune cell cytotoxicity toward NSCLC was significantly higher for intratumor NKs compared to other lymphocyte populations in both ex vivo and in vivo immunocompetent mouse models.

In this study, we demonstrated that BETi, through the specific inhibition of the epigenetic reader BRD4, down-regulated the expression of inhibitory receptors and molecules that are involved in NK cell regulatory mechanisms, self-tolerance, and tumor immune escape. These data are in line with other studies showing that ICs are expressed and fully functional in NK cells, besides T lymphocytes, further supporting the leading role of NK cells as primary targets of IC inhibitors in the tumor immune landscape[44,45].

In our RNA-seq analysis, we identified a specific signature of 80 genes associated with NK exhaustion, which included classical and non-classical ICs, iKIRs, and IC-related molecules. The entire signature was down-regulated by BETi, further supporting the rationale of using BETi to maximize immunotherapy response.

PD-1/PD-L1 axis is a primary target of BRD4 inhibition as it was already reported[25,42]. Here, we showed that BRD4 controls a widespread panel of other inhibitory ICs, including CTLA4 and the so-called next-generation IC molecules, such as LAG3, TIM3 and TIGIT, which are currently targets in clinical trials[46].

Among BRD4 targets, we also identified CEACAM1, which was reported to inhibit NK-mediated cytolysis of tumor cells[47,48] and KLRG1, a negative regulator of CD8+ lymphocytes and NK cells[49].

Notably, CLECL2D and CLEC12A are C-type lectin receptors encoded by NK cells with a critical role in self and non-self-recognition[50]. Their enhanced expression on tumor cells and NK cells suppresses the activity of lymphocytes and other TME immune cells, fuelling a negative regulatory circuit that triggers tumor immune escape[51].

Besides ICs, NK activation is tightly regulated by HLA-class I-specific iKIRs which allow self- and non-self-discrimination. We demonstrated that BRD4 inhibition was sufficient to down-regulate several iKIR, including KIR2DL1, KIR2DL3, and KIR3DL3[52]. KIR2DL4 is an atypical KIR that brings an arginine-tyrosine activation motif in the transmembrane region and an immunoreceptor tyrosine-based inhibitory motif (ITIM) in the cytoplasmic tail, suggesting to be either activating or inhibitory receptor[53]. Still, KIR2DL4 interaction with HLA-G seems to increase PD-1/PD-L1 expression and thus increase NK cell exhaustion[54]. Of note, we detected a down-regulation of KIR3DX1, which is a pseudogene encoding for an Ig-like receptor and its function is currently unknown, but it was reported to be enhanced in NK cells from responder patients to anti-PD-1 blockade[55].

CEACAM3 is mainly expressed in granulocytes and seems to have a role in controlling phagocytosis[56], whereas its role in NK effector properties has been poorly investigated. We found that CEACAM3 is one of the main targets of BRD4 in NK cells, supporting future studies to decipher its role in NK immunity.

ICs and iKIRs are important mediators of the stability of the immunological synapses that are established between immune and tumor cells[27]. These complex molecular interactions engaged by effector immune cells on the surface of target tumor cells are critical to obtaining a proficient immune activation. Indeed, the formation of stable immunological synapses leads to a strong engagement of immune cells, promoting their effector properties. Several therapies have been developed to achieve a direct manipulation of immunological synapses, including IC blockade. In this work, we demonstrated that the BETi-mediated down-regulation of ICs and iKIRs had a relevant impact on the generation of immunological synapses, deeply affecting the efficiency of NK target tumor cell recognition.

The opportunity of hitting a large set of inhibitory receptors by pharmacological inhibition or by genomic silencing of BRD4 can impact the future design of IC blockade protocols. Indeed, BETi administration can overcome those IC blockade resistance mechanisms that rely on high levels of alternative ICs or co-inhibitory receptors[33]. In this work, we demonstrated that BETi had a superior

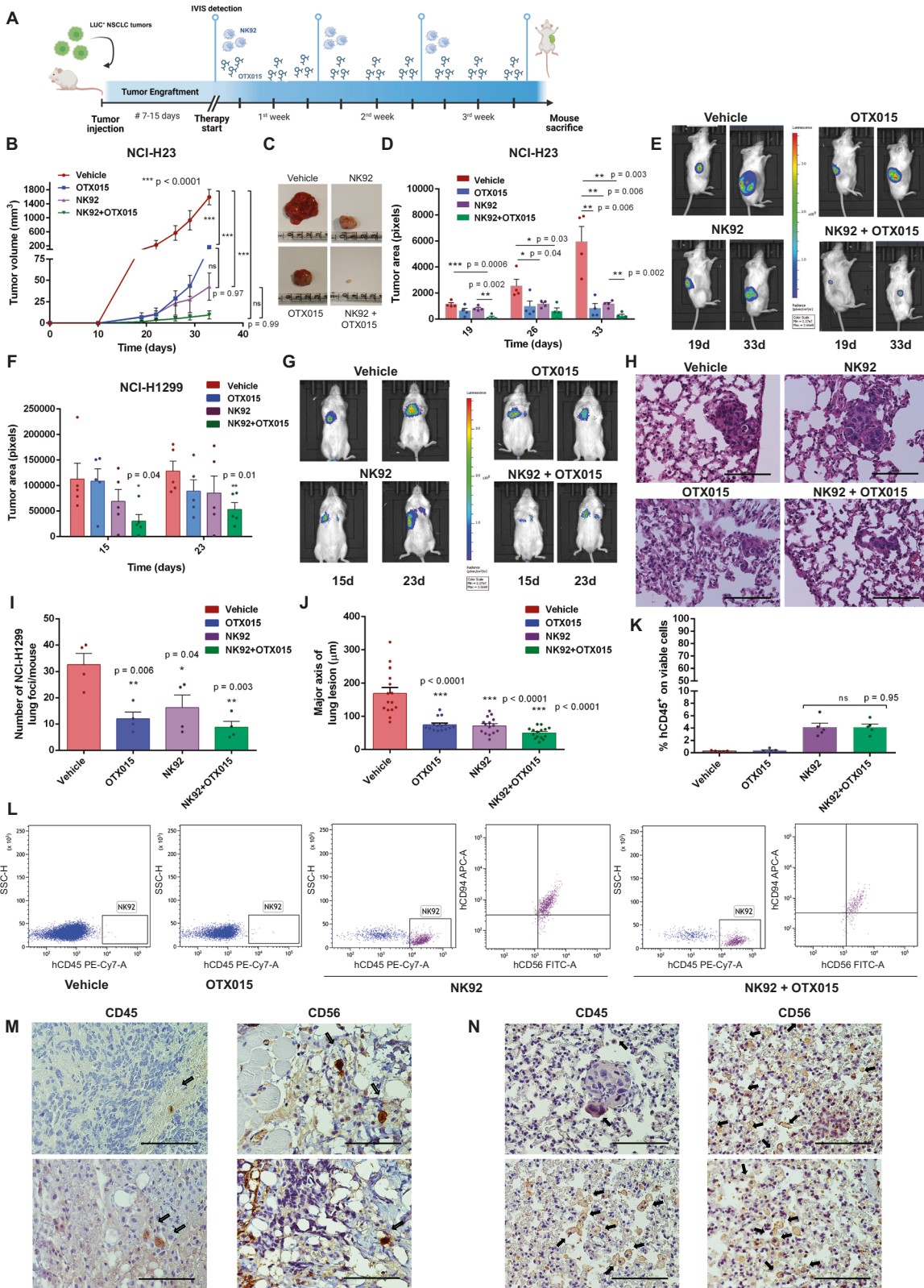

ability to enhance NK cytotoxicity compared to single-agent IC inhibitors. According to our observations that BETi exerted a concerted down-regulation of multiple ICs and iKIRs, the addition of IC inhibitors did not improve BETi's overall anti-tumor effect. These data further supported that IC inhibition is a key mechanism in the observed BETi-mediated regulation of NK activity toward NSCLC.

We demonstrated the direct transcriptional cooperation of BRD4 and SMAD3 in controlling the expression of NK exhaustion genes and immunological synapse regulation (Fig. 8). We showed that BRD4 and SMAD3 belonged to a common transcription complex and co-occupy the same genomic regions. This transcriptional cooperation was not limited to the NK context, but a genomic overlap between BRD4 and

**Fig. 6 | BETi maximize the efficacy of an adoptive NK cell therapy in mice.**
**A** Overview of xenograft experiments. Therapy was started when tumor growth was detected by bioluminescence signal (IVIS), 13 or 7 days after tumor injection for NCI-H23 or NCI-H1299, respectively. Mice were euthanized after 3 or 2 weeks from the start of the therapy for NCI-H23 or NCI-H1299, respectively. Image was created by Biorender.com; **B–E** NCI-H23-LUC subcutaneous growth was monitored with digital calliper (**B-C**).Time is expressed as days from tumor injection (**B**, $n = 5$ mice per group, two-way ANOVA with multiple comparison (Tukey test) was applied). IVIS system monitored tumor spreading and invasion (**D–E**). The combination of NK92 administration and OTX015 (5 mg/Kg) was more efficient in restraining tumor growth compared to single treatments or vehicle (**D**, $n = 4$ mice per group); **F, G** IVIS system was applied to monitor NCI-H1299-LUC orthotopic lung engraftment. Time is expressed as days from tumor injection. The combined treatment displayed a higher anti-tumor effect compared to other arms ($n = 5$ mice per group); **H–J** Tumor lung lesions were analysed by pathological assessment of H&E slides (40x magnification, scale bar 100 μm, **H**). NCI-H1299-LUC lung foci number (**I**, $n = 4$ mice per group) and the major axis of lesions (**J**, $n = 15$ lesions per group) were reduced in mice administered with NK92 and OTX015; **K, L** NK92 infiltration in tumors from NCI-H23 xenografts was quantified by flow cytometry (**K**). NK92 tumor recruitment was not affected by OTX015 (results are mean of % hCD45+ cells on total viable cells, $n = 5$ mice per group, **L**); **M, N** Representative IHC images of NCI-H23 tumors (**M**) or lungs collected from NCI-H1299 xenografts (**N**). For each specimen, two slides were stained with anti-hCD45 or anti-hCD56 to spatially identify NK92 infiltration (40 x magnification, scale bar 100 μm). Black arrows highlight NK92. IHC staining was performed on 5 mice per group. All data in this figure are shown as mean ± SEM. If not otherwise specified, two-side Student's $t$-test was applied. If not specified, ***$p \leq 0.0001$, ns: not significant. Source data are provided as a Source Data file.

SMAD3 peaks at the TSS of protein-coding genes was corroborated by ChIP-seq data analysis in K562 cells. Still, the transcriptional program controlled by the BRD4/SMAD3 complex emerged to be highly cell-dependent and, in the context of NKs, strictly involved in the regulation of exhaustion markers.

BRD4 and SMAD3 physical interaction was previously identified in myofibroblasts, in which their transcriptional complex mediates TGF-β response[57], but it was never reported in NK cells. Of note, we found SMAD3 to preferentially bind either 3'UTR or TSS regions of target genes, suggesting a three-dimensional chromatin reorganization triggered by BRD4-SMAD3 interaction.

In this study, we demonstrated that the functional cooperation of SMAD3 with BRD4 was required to properly control IC and iKIR expression in NKs. Indeed, SMAD3^KD in NK92 cells and in patient-derived NKs significantly impaired the expression of these molecules. Conversely, SMAD3 induction upon TGF-β stimulation rescued BETi-induced down-regulation of many of IC and iKIR molecules, in line with the fact that SMAD3 cooperated with BRD4 in their regulation.

Overall, we speculated that BRD4 modulated SMAD3 transcriptional function through a multi-layered mechanism. The pharmacological inhibition of BRD4 was sufficient to prevent/displace the BRD4-SMAD3 complex occupancy from target genomic regions. Moreover, SMAD3 was a direct transcriptional target of BRD4 and BRD4 inhibition was sufficient to reduce its expression in NK cells.

This is not surprising evidence, since BRD4 is known to cooperate with other TFs in transcriptional complexes and concomitantly controls the expression of its co-operators, such as we previously demonstrated for YAP and TAZ in NSCLC cells[34].

Other independent studies have identified the inhibitory role of SMAD3 on NK cells[58,59]. Silencing of SMAD3 enhanced NK cell development in mouse models and stimulated NK-dependent anti-tumor immunity. In this context, SMAD3 repressed transcription of IFN-γ regulating E4BP4 in NK cells[59]. SMAD3 is also a known effector of TGF-β signaling, a major immune suppressive pathway that restrains NK activation through the down-regulation of NK stimulatory receptors[32,60]. We demonstrated that TGF-β stimulation was sufficient to induce the expression of several ICs and iKIRs in NKs, confirming that SMAD3 was a required upstream regulator. We may speculate that NK exhaustion in the TME is a broad consequence of the elevated levels of TGF-β, which drives the SMAD3-dependent transcriptional program in NK cells. This effect can be only partially reverted by BETi treatment with a general rescue of the NK anti-tumor potential.

In this framework, our data provide an additional piece of evidence and highlight an underscored SMAD3 function in NK cells, demonstrating its direct engagement in IC and immunological synapse regulation. Indeed, we observed that most of the genes belonging to our NK exhaustion signature were controlled by SMAD3/BRD4, pointing to this axis as a pivotal mediator of the BETi stimulatory effect. Still, SMAD3 silencing was not sufficient to fully recapitulate the improved NK activation achieved through BETi, therefore indicating the existence of other SMAD3-independent effects.

Among the top-scoring BETi down-regulated pathways, we detected IL-6 signaling which is known to repress NK activation[61]. Besides, BETi up-regulated several genes involved in calcium-related pathways and microtubule assembly, indicating a potential regulation of intracellular vesicle trafficking and release of cytotoxic granules. Other works highlighted the use of BETi to elicit NK cytotoxicity in several malignancies by identifying other mechanisms, including increased NKG2D expression[39]. Conversely, other studies reported a contradictory role of BETi in impairing NK cytolytic activity and inflammatory potential, although these observations were made on peripheral blood mononuclear cells (PBMC)-derived NK cells from healthy subjects[29,62]. The immunoregulatory role of BET proteins appears strictly cell- and context-dependent and may include concerted pro- and anti-inflammatory effects depending on the specific microenvironment and stimulatory signals[38]. Future studies will be required to fully understand the complexity of the epigenetic regulation orchestrated by BRD4 and other BET proteins in NK cells.

Of note, we found that BETi induced a concomitant down-regulation of ICs and HLA class I molecules in NSCLC cells. The effect was not limited to the well-known PD-L1 down-regulation[25] but included the inhibition of PD-L2, LSGAL9, HVEM and other IC tumor ligands. The absence of HLA class I molecules on tumor cells is associated with an increased ability of NKs to recognize them as non-self-targets. Taken together, the concomitant down-regulation of HLA class I molecules and ICs further triggered NK activation toward NSCLC cells. These results may indicate that in this context BETi effects are overall synergistic with NK administration because the drugs regulated tumor cell-autonomous effects that led to a more efficient NK anti-tumor recognition and response. Future studies will be required to address this hypothesis.

The use of NK cells in adoptive cell therapies has been applied to different cancer contexts and with different protocols, including both autologous and allogeneic cell transplantations. Besides, NK cells can be engineered to produce Chimeric Antigen Receptor (CAR)-NK cells or combined with target-specific antibodies to stimulate antibody-dependent cell-mediated cytotoxicity[63,64]. All these strategies have showed encouraging results but also drawbacks, especially in solid tumors. Therefore, the combined treatment of BETi with target-specific antibodies, such as cetuximab, may be a valuable tool to enhance adoptive NK cell therapies in lung cancer patients.

Collectively, our work provides relevant implications for the management of immunotherapy protocols due to the identification of molecular targets (BRD4 and SMAD3), whose inhibition has been shown to specifically enhance NK cytotoxicity and reduce NK exhaustion.

Our results showed a proof-of-principle that the concomitant administration of the epigenetic drugs BETi can significantly increase

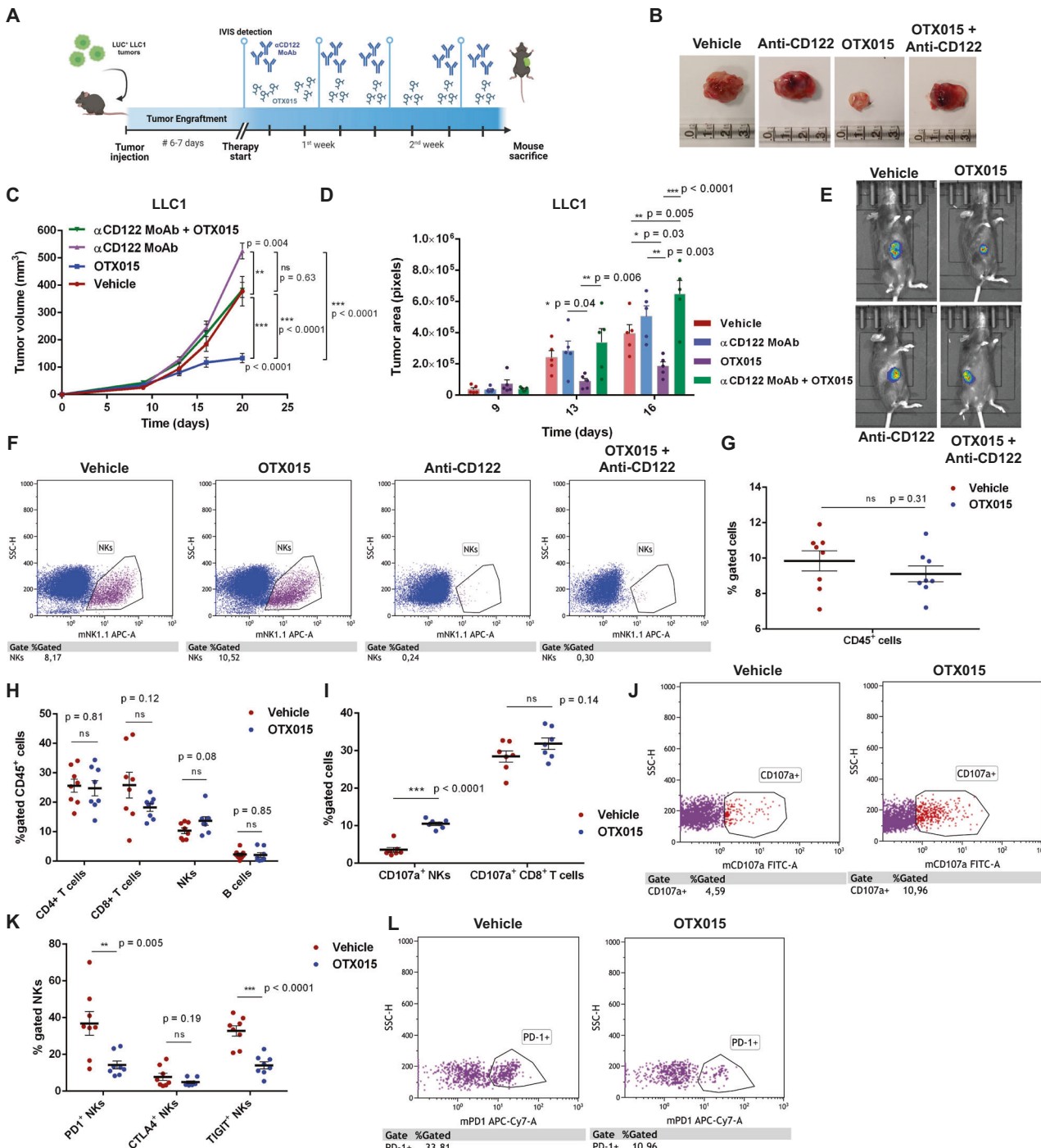

**Fig. 7 | NKs are required to maximize BETi anti-tumor activity in NSCLC syngenic models. A** Overview and timeline of the syngenic models developed using the murine LLC1-LUC1 cell line. After subcutaneous engraftment (1-week post tumor injection), mice were randomized in different experimental groups ($n = 8$ for each arm) and administered with vehicle, OTX015 (5 mg/Kg), anti-CD122 MoAb (300 μg/mouse) or their combination. Mice were euthanized 21 days after tumor injection. The image was created by Biorender.com; **B** Representative images of the different tumor growth at mouse sacrifice; **C–E** Tumor growth was monitored with a digital caliper (**C**, $n = 6$ mice per group, two-way ANOVA with multiple comparison (Tukey test) was applied) and by quantifying the in vivo tumor bioluminescence with IVIS system (**D**, **E**, $n = 5$ mice per group, two-way ANOVA with multiple comparison (Tukey test) was applied). Time is expressed as days from tumor injection; **F** The efficiency of NK cell depletion induced by anti-CD122 MoAb was confirmed in tumor specimens by flow cytometry analysis of murine NKs (CD45$^+$NK1.1$^+$cells); **G** Immune infiltration (CD45$^+$ cells) in mouse tumors was assessed by flow

cytometry analysis. No differences were observed following OTX015 administration ($n = 8$ mice per group); **H** Quantitation of different TIL subsets comparing OTX015- to vehicle-treated mice by flow cytometry analysis. The main detected populations were T cells (CD4$^+$ and CD8$^+$ subsets) and NKs, whereas B cells (CD19$^+$) were poorly represented in these tumors. BETi did not significantly affect the percentage of infiltration/recruitment of different cell types ($n = 8$ mice per group); **I, J** The degree of activation of cytotoxic populations, CD8 T lymphocytes and NKs, was measured by CD107a surface expression in mouse tumors by flow cytometry. OTX015 specifically increased NK cell degranulation ($n = 7$ mice per group); **K, L** IC expression was evaluated by flow cytometry on tumor-infiltrated NKs and detected down-regulated by OTX015 ($n = 8$ mice per group, data are shown as mean ± SEM). All data in the figure are shown as mean ± SEM. If not otherwise specified, two-side Student's $t$-test was applied. ns: not significant. Source data are provided as a Source Data file.

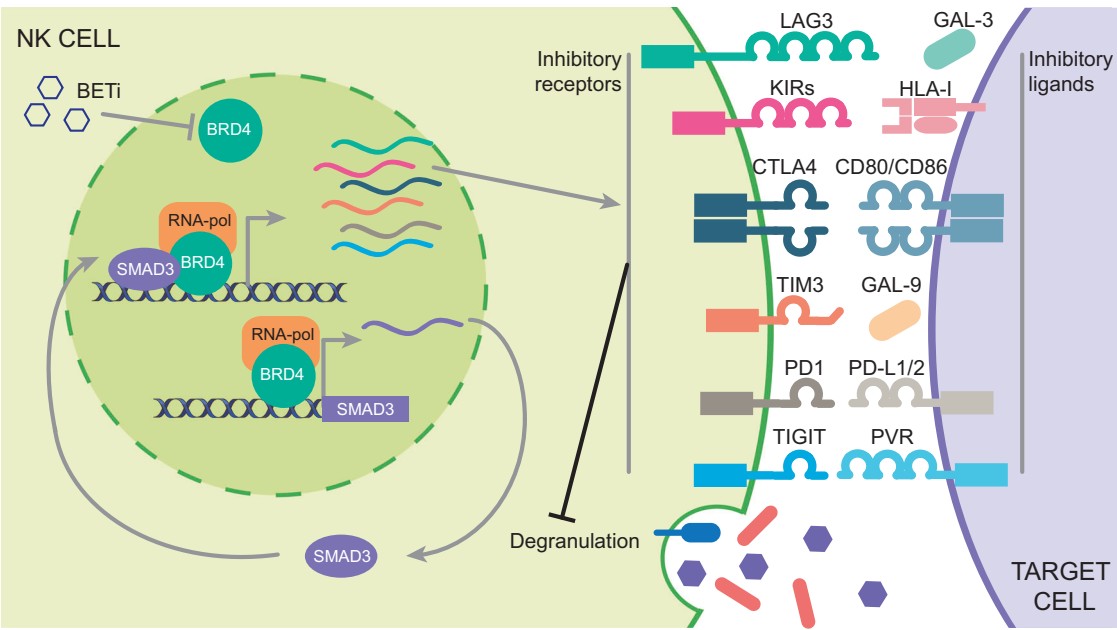

**Fig. 8 | Graphical Illustration of the multilayer regulation orchestrated by BRD4 on SMAD3 expression and activity in NK cells.** SMAD3 is a direct transcriptional target of BRD4 in NK cells. In addition, BRD4 interacts with SMAD3 and their concerted activity regulates the expression of NK inhibitory receptors. This limits NK cytotoxicity by restraining the recognition of target cells. BETi restrain the transcriptional activity of both BRD4 and SMAD3, inducing a down-regulation of NK inhibitory molecules, including several ICs and iKIRs. The graph illustrates the main inhibitory receptors that are repressed by BETi and that we found under the control of BRD4/SMAD3 cooperation, together with the counterpart interacting ligands expressed by tumor cells.

the efficacy of adoptive NK cell therapies in NSCLC mouse models and ex-vivo patient-derived 3D models. Moreover, we demonstrated that NKs were relevant immune cell players in NSCLC TME and their functionality was required to obtain an efficient anti-tumor response promoted by BETi in immunocompetent mouse models.

The inhibition of BRD4 and SMAD3 through BETi or genomic editing can be valuable therapeutic strategies to exploit the anti-tumor potential of NK cells in adoptive NK/CAR-NK cell therapies or, alternatively, to rescue/enhance autologous NK activity in TME of NSCLC patients.

## Methods

### Ethics statement

The study was approved by the local ethical committee (Comitato Etico dell'Area Vasta Emilia Nord (AVEN) - Reggio Emilia district, Italy, authorization code 196/2017) and conformed to all Institutional and National laws and guidelines. All patients were informed and signed a written consensus.

### Patient specimens

A total of 53 patients with new NSCLC diagnoses at the Azienda USL-IRCCS of Reggio Emilia (Italy) from 2018 to 2023 were included in the study. Supplementary Data 1 summarized patient clinicopathological features. Our cohort was composed of 35 cases of AD, 16 cases of SQ, and 2 cases of Typical Carcinoids.

Briefly, fresh surgical tumor specimens were enzymatically digested with Liberase™ DH Research grade (Sigma-Aldrich, St. Louis, Missouri, USA, 0.26 IU/ml) for 1 h at 37 °C. The cell suspension was then filtered with a 100 µm cell strainer to remove tissue debris and further filtered with a 40 µm strainer.

The tumor fraction left on the 40 µm strainer was collected to isolate the cancer-tissue originated spheroids (CTOS), three-dimensional tumor aggregates with a diameter of 40–100 µm[26]. CTOS were cultured in DMEM-F12 medium (Thermo Scientific, Waltham, Massachusetts, USA) containing EGF (PeproTech, Cranbury, NJ, USA, 20 ng/mL), bFGF (PeproTech, 20 ng/mL),

B27 supplement (Gibco, Life Technologies, 1x), and heparin sodium (Sigma-Aldrich, 0.6 IU/mL) or embedded in 100% Matrigel Basement Membrane (Corning, Glandale, Arizona, USA) and overlaid by the medium.

The single-cell solution (<40 µm tumor fraction) was further purified by Ficoll gradient (Histopaque®-1077, Sigma-Aldrich) at 700 x g for 20 min. The ring of cells in the interphase between the Ficoll and the solution, corresponding to mononuclear tumor-infiltrating leukocytes (TILs), was collected through a sterile pipette. Natural Killer (NK) cells were further magnetically purified from this fraction using the NK cell isolation kit (#130-092–657 Miltenyi Biotech, Bergish Gladbach, Germany). CD4⁺ and CD8⁺ T-lymphocytes were magnetically purified from TIL fraction using CD4 human microbeads (#130-045-101 Miltenyi Biotech) or CD8 human microbeads (#130-045-201 Miltenyi Biotech), respectively. TIL and NK cell purity was checked by flow cytometry analysis (Suppl. Fig. 1A, F). TILs, CD4⁺, CD8⁺ or NK cells were short-term cultured in RPMI medium (Thermo Fisher Scientific) added with Pen/Strep (Euroclone, Milan, Italy, 1%), Fetal Bovine Serum (FBS, Euroclone, 10%), and Recombinant Human IL-2 (Prepotech, London, UK, 20 IU/ml).

### Cell cultures and treatments

NCI-H23, NCI-H1299, and NCI-H1975 NSCLC cell lines were obtained from Dr. Massimo Broggini (IRCCS-Istituto di Ricerche Farmacologiche Mario Negri, Milan, Italy). Lewis Lung carcinoma (LLC1) cell line was obtained from IFOM-IEO Campus, Milan, Italy. HEK293T (ATCC CRL-3216) and NK92® (ATCC CRL-2407) cell line was purchased from ATCC (LGC Standards, Sesto S.Giovanni, Italy). Human NSCLC and HEK293T cell lines were sub-cultured in RPMI-1640 medium (Thermo Fisher Scientific) added with FBS (Euroclone, 10%) and antibiotics at 37 °C/5% CO₂. NK92 cells were cultured according to the manufacturer's instructions[28]. LLC1 cells were sub-cultured in DMEM medium (Thermo Fisher Scientific) added with FBS (Euroclone, 10%) and antibiotics at 37 °C/5% CO₂. All cell lines have been authenticated through SNP or STR profiling by Multiplexion Gmbh (Heidelberg, Germany, last authentication was performed on 29th June 2023) and

were routinely checked (i.e. every 2 weeks) to be negative for myco-plasma contamination.

If not otherwise specified, cells were treated for 24 h with 1 μM JQ1 (#HY-13030, MedChem Express, Monmouth Junction, NJ, USA) or 1 μM OTX-015 (Birabresib, #HY-15743, MedChem Express) previously dissolved in DMSO and further diluted in complete medium. DMSO alone was used as treatment control (vehicle). BETi concentration was chosen based on our previous studies performed on NSCLC cell lines in which this dosage corresponded to IC50 at 72 h[34]. 1 μM of BETi was sufficient to inhibit the expression of BRD4 targets after 24 h with a reduction >50% in both NK92 and NSCLC cells (Suppl. Fig. 5F−G, Suppl. Fig. 9F−I).

Monoclonal antibodies targeting PD-1 (Nivolumab, 15 μg/ml, #HY-P9903, MedChem Express), CTLA4 (Ipilimumab, 5 μg/ml, #HY-P9901, MedChem Express) or TIGIT (Vibostolimab, 10 μg/ml, #HY-P99202, Medchem Express) were added directly to the culture media.

Recombinant human TGF-β was administered to NK92 cells at the concentration of 5ng/ml (#100-21, Peprotech).

### Immune cell stimulation with PMA/Ionomycin
TILs, NKs or NK92 cells were stimulated for 6 h with standard PMA (Sigma-Aldrich, 50ng/ml) and Ionomycin (Sigma-Aldrich, 1 μg/ml) treatment. Golgi Plug (#555029, BD), containing Brefeldin A, was added to culture media to prevent IFN- γ release. During stimulation, cell media contained either BETi or vehicle and were all supplemented with IL-2 (20 IU/ml). For the intracellular staining of IFN-γ, the Fixation/Permeabilization kit (#554714, BD) was used according to the manufacturer's instructions.

### Immune cell assays
For 2D co-cultures, NCI-H23, NCI-H1299 and NCI-H1975 were seeded in 96-well plates (2000 cells/well) the day before. For 3D co-cultures, spheroids of NCI-H23, NCI-H1299 and HCI-H1975 were generated in 96-well Ultra-Low attachment plates (Corning) by seeding 500−2000 cells/well and centrifuging at 130 x g for 15 min and incubated for 48 h at 37 °C/5%CO2. Patient-derived CTOS were seeded on Geltrex™ membrane matrix (Gibco, Thermo Fisher Scientific) pre-coated 24-well plates the day before the co-culture. BETi treatment (1 μM JQ1 or OTX015) or vehicle was also added to immune cells (NK92 or primary TILs/NKs) the day before starting of immune cell assays with tumor cells. After 24 h of BETi or vehicle pre-treatment, NK92 or primary TILs/NKs were stained with Incucyte® Cytolight Rapid dyes (#4705 green or #4706 red, Sartorius, Gottinga, Germany) following the manufacturer's instructions. Immune red-stained cells were then added to tumor 2D/3D cultures (500 cells/well) in presence of IL-2 (20 IU/ml) and IL-15 (Peprotech, 5 ng/ml). Tumor cell growth was monitored using Incucyte® S3 live cell imaging system (Essen Biosciences Inc, Ann Arbor, MI, USA) and quantified as confluence cell area for 2D cultures or spheroid brightfield area for 3D cultures by Incucyte® software (2022B version, Essen Biosciences Inc). Immunological synapses were quantified in 3D co-cultures as red fluorescence intensity within the brightfield area of the spheroid by Incucyte® software. In 2D co-cultures, immunological synapses were counted as red objects within the tumor cell phase area and normalized to time 0 using Incucyte® software. Tumor cell cytotoxicity was quantified by adding to co-culture media Incucyte® Cytotox Green dye (#4633, Sartorius, 250 nM) and measuring mean green fluorescence by Incucyte® software.

Tumor apoptosis and immune cell activation were evaluated 24 h after the starting of the co-cultures between immune cells and NSCLC cells (ratio E:T 1:8) by applying flow cytometry analysis.

### Flow cytometry analysis
Cell samples were resuspended in cold FACs Buffer (1 x PBS, 5mM EDTA, 2% BSA) and stained with the antibodies listed in Supplementary Table 1. 7AAD staining (#559925, BD, Franklin Lakes, NJ, USA) was used to exclude dead cells. FcR Blocking Reagent was used to prevent unspecific antibody binding (Miltenyi Biotech). Apoptosis analysis was performed with the PE Annexin V Apoptosis Detection kit (#559763, BD) and according to the manufacturer's instructions. The intracellular staining of IFN-γ was assessed applying the Fixation/Permeabilization kit (#554714, BD). Samples were acquired with FACS Canto II (BD) and BD FACSDIVA v 8 Software (BD). Data were analysed with Kaluza software v2.1 (Beckman Coulter, Brea, CA, USA).

### RNA-sequencing (RNA-seq)
Total RNA was extracted with the RNeasy Mini Kit (Qiagen, Hilden, Germany) and on-column DNase treatment according to the manufacturer's instructions. RNA-seq libraries were obtained starting from 500 ng of total RNA following TruSeq Stranded Total RNA preparation protocol (Illumina, San Diego, CA, USA). Sequencing was performed through Illumina NextSeq high-output cartridge (double-stranded, reads length 75 bp−2 × 75). Quality check and data analysis were performed as previously described[65].

### Bioinformatic analyses
Sequencing quality was assessed using the FastQCv0.11.8 software (www.bioinformatics.babraham.ac.uk/projects/fastqc/), showing on average a Phred score per base >34 in each sample. Raw sequences were aligned to the human reference transcriptome (GRCh38, Gencode release 35) using STAR version 2.7 and gene abundances were estimated with RSEM algorithm (v.1.3.1). Differential expression analysis was performed using DESeq2 R package (R software v4.1.0), considering an adjusted $p$-value < 0.05 and excluding genes with low read counts. Significantly deregulated genes (DEGs) underwent enrichment analysis, performed on Gene Ontology Biological Processes, KEGG, and Reactome pathways databases by enrichR R package using a significance threshold of 0.05 on adjusted $p$-value. The protein-protein interaction network of top-scoring DEGs was analysed using the STRING tool v11.5 (http://www.string-db.org/).

Promoters of 80 selected DEGs were extracted from TxDb.Hsapiens.UCSC.hg38.knownGene annotation package (v 3.3.0) through GenomicFeatures R package (v 1.46.5) considering a window of [−2000bp, +400bp] around Transcription Starting Site (TSS). Bedtools getfasta was used to get FASTA sequences of promoters[66]. Selected promoter regions were scanned for Transcription Factor (TF) DNA binding motifs by FIMO algorithm[67] (source MEME suit), using HOCOMOCO and JASPAR as reference motifs databases and adjusted $p$-value < 0.05 as significance threshold to filter and rank the resulting motif occurrences.

### Gene silencing
NK92 or patient-derived NK cells were transfected with Lipofectamine RNAiMax Reagent (Life Technologies, Monza, Italy) and 50nM Silencer™ Select siRNAs (a mix of s23901 and s23902 for BRD4; s8401 or s8402 for SMAD3; 4390847 for the negative control, Thermo Fisher Scientific).

### Cytoplasm/nucleus fractioning
NK92 cells were treated with 5 ng/ml TGF-β for 6 h to induce SMAD3 nuclear localization and activation. Cells were then harvested and washed twice with cold 1 x PBS. The cell pellet was resuspended in a hypotonic Buffer (10 mM HEPES pH7.9; 10 mM KCl; 1.5 mM MgCl2; 0.05% NP40) supplemented with the protease inhibitors for 5 min on ice. After sample centrifugation at 3000 x g for 2 min at 4 °C, the supernatant was collected as the cytoplasmic fraction. The pellet was then resuspended in a Nuclear Lysis Buffer (50 mM Tris pH7.4; 150 mM NaCl; 1% Triton-x100; 1mM EDTA) supplemented with the protease inhibitors for 30 min at 4 °C. Samples were then centrifuged at 14000 x g for 10 min and the supernatant was collected as the nuclear

fraction. Fractions were quantified with Bradford Protein Assay (Bio-Rad, Hercules, CA, USA).

## Western blotting

The total cell lysate was obtained with PLB Buffer (Promega, Madison, Wisconsin, USA, 1x) added with protease inhibitor cocktail (Sigma-Aldrich, 1x). Soluble proteins were separated from debris by 10 min centrifugation at 14000 x g and quantified with Bradford Protein Assay. Western blot analysis was performed as previously described[34]. The following antibodies were used, following the manufacturer's instructions: rabbit anti-SMAD3 (#9523, Cell Signaling Technology, Danvers, MA, USA, 1:1000), rabbit anti-BRD4 (#128874, Abcam, Cambridge, UK, 1:1000), mouse anti-beta actin (#A2228, Sigma-Aldrich, 1:5000), anti-Rpb1 NTD (RNA Pol II, #14958S, Cell Signaling Technology, 1:1000), anti-α-tubulin (#sc8035, Santa Cruz Biotechnologies, Dallas Texas, USA, 1:1000), Amersham ECL™ donkey Anti-Rabbit IgG Horseradish Peroxidase-conjugated antibody (#NA934V, Cytiva, Marlborough, MA, USA, 1:500), Amersham ECL™ sheep anti-mouse IgG Horseradish Peroxidase-conjugated antibody (#NXA931V, Cytiva, 1:5000). Images were acquired with Image Lab v 5.2 Software (Biorad). Densitometric analysis was performed using ImageJ software v1.54 (source https://imagej.nih.gov/ij/). Uncropped blots were provided in the Source Data File and in the Supplementary Information File.

## Quantitative PCR (RT-qPCR)

Total RNA was extracted with the RNeasy Mini Kit (Qiagen, Hilden, Germany) and on-column DNase treatment according to the manufacturer's instructions. Total RNA was retrotranscribed using the iScript™ cDNA kit (Bio-Rad). Quantitative real time-PCR (qRT-PCR) was conducted using Sso Fast EvaGreen Super Mix (Bio-Rad) in the CFX96 Real-Time PCR Detection System (Bio-Rad) with the Bio-Rad CFX Manager v 3.1 Software (Biorad). GUSB and Cyclophilin A were used as reference genes to normalize gene expression that was calculated with the delta-delta Ct method ($2^{-\Delta\Delta Ct}$). Primer sequences are shown in Supplementary Data 2.

## Enzyme-Linked ImmunoSorbent Assay (ELISA)

Cell supernatant of co-cultures between patient-derived NKs and NSCLC cells (ratio E:T 1:8) was collected after 24 h, centrifuged at 2000 x g for 5 min to remove cell debris, and stored at −20 °C. Human IFN-γ quantitation was performed using Quantikine® ELISA kit (#DIF50C, R&D Systems, Minneapolis, Minnesota USA) and the following manufacturer's instructions.

## Chromatin Immunoprecipitation (ChIP)

ChIP analysis was performed on NK92 cells as previously described[35]. Briefly, NK92 cells were cross-linked for 15 min with 1% formaldehyde, lysed, and sonicated for 10 cycles (30 sec ON/30 sec OFF each cycle) with Bioruptor Pico Sonicator (Diagenode, Denville, NJ, USA). Chromatin was precipitated using Dynabeads Protein G magnetic beads (Thermo Fisher Scientific) and rabbit anti-BRD4 (#A301-985A100, Thermo Fisher Scientific, 5 μg/30 x 10^6 cells) or rabbit anti-SMAD3 (#9523, Cell Signaling Technology, 1:50) or equivalent amount of Normal Rabbit IgG isotype control (#2729, Cell Signaling Technology). A fraction equal to 0.25% of total chromatin was used as input. After reverse cross-linking and protease K treatment, genomic DNA was purified with the the QIAquick® PCR Purification kit (#28106, Qiagen). Primer sequences that were used to amplify target genomic regions are listed in Supplementary Data 3. An intergenic sequence was used as a negative control as previously described[35]. Actin B promoter region was used as a positive control for BRD4 binding, whereas *E4BP4* 3'UTR was the positive control for SMAD3 binding[58]. Each qPCR value was normalized over the appropriate input control and reported in graphs as % of input.

## Bioinformatic analyses of ChIP-seq public datasets

The following ChIP-seq public datasets were analysed: GSE156423 ChIP-seq for BRD4 in NKs isolated from peripheral blood of two healthy donors[29], GSE101225 (ENCODE project n°ENCSR583ACG) ChIP-seq for BRD4 in K562 cell line, GSE231137 (ENCODE project n°ENCSR140GLO) ChIP-seq for SMAD3 in K562 cell line[30]. Raw Fastq files of the GSE156423 dataset were downloaded from the GEO database through sra toolkit (source GitHub), then processed as previously described[68]. All processed data of GSE101225 (ENCSR583ACG) and GSE231137 (ENCSR140GLO) datasets were downloaded from ENCODE platform[30].

To detect BRD4 counts, we selected 2 kbp windows around the center of ChIP target regions (amplified by primers listed in Supplementary Data 3), then multiBAM summary from deepTools v3.5.1[69] was applied on bam files to extract BRD4 counts over these genomic regions.

The overlap between BRD4 and SMAD3 was evaluated by applying the bedtools intersect function (bedtools suite, source GitHub) on conservative IDR thresholded peaks considering a minimum overlap of 1 bp. ChIP-seq metaprofile was generated with deepTools v3.5.1 (https://deeptools.readthedocs.io/en/develop/index.html), considering fold-change over control bigwig files of both replicates. Enrichment was calculated over the conservative IDR thresholded peaks of SMAD3, scaled to the average peak length, considering 3 kbp up- and downstream of the peaks. Promoter regions, defined as 1.5 kbp up- and down-stream of the TSS, of protein-coding genes were extracted from Gencode v44 basic annotation through GenomicFeatures R package. Mitochondrial genes were excluded. BRD4 and SMAD3 counts at selected promoters were extracted by multiBAM summary from deepTools (v3.5.1)[69], summed to collapse replicates, then shown as a scatter plot. BRD4 and SMAD3 correlation on selected promoters was quantified using Spearman's correlation coefficient.

## Co-Immunoprecipitation (Co-IP)

NK92 cells were washed twice with cold PBS and resuspended in hypotonic buffer (10 mM HEPES pH 7.9; 10 mM KCl; 1.5 mM MgCl$_2$; 0.05% NP40) supplemented with protease inhibitors for 5 min on ice. After centrifugation at 400 x g for 2 min at 4 °C, the supernatant was collected as the cytoplasmic fraction. The pellet was then resuspended in nuclear lysis buffer (50 mM Tris pH 7.4; 150 mM NaCl; 1% Triton-x100; 1 mM EDTA) supplemented with protease inhibitors for 30 min at 4 °C. Samples were then centrifuged at 14000 x g for 10 min and the supernatant was collected as the nuclear fraction. Fractions were quantified with Bradford Protein Assay. 1 mg of total nuclear lysate was kept as input. Sepharose beads (Protein A Sepharose®CL4B, Sigma-Aldrich) were pre-coated with rabbit anti-BRD4 antibody (#A301-985A100, Thermo Fisher Scientific, 2 μg/1mg of cell lysate) or Normal Rabbit IgG isotype antibody (#2729, Cell Signaling Technology, 2 μg/1mg of cell lysate) for 1 h at 4 °C. The nuclear lysate was pre-cleared with uncoated sepharose beads for 1 h at 4 °C. After beads centrifugation at 400 x g for 5 min at 4 °C, the supernatant of pre-coated beads was discarded, whereas the supernatant of pre-cleared nuclear lysate was collected. The pre-cleared supernatant was equally divided into two tubes, each one added with antibody coated-beads for αBRDA or IgG isotype control. After overnight incubation at 4 °C, beads were washed with cold Tris-Buffered Saline (50 mM Tris pH7.4, 150 mM NaCl). Each sample was further divided into two tubes added with Laemmli Sample Buffer (Biorad) and beta-mercaptoethanol for BRD4 detection or Laemmli Sample Buffer without beta-mercaptoethanol for SMAD3 and analysed by Western Blotting.

## Luciferase expressing cells

NCI-H23, NCI-H1299 and LLC1 cells were engineered for firefly luciferase expression to obtain NCI-H23-LUC, NCI-H1299-LUC, and LLC1-

LUC, respectively. HEK293T cells were transfected with Lipofecta-mine2000 Reagent (Life Technologies), 3rd generation lentivirus packaging system, and the Luciferase stable expressing plasmid (pLenti CMV Puro LUC, w168-1, Addgene #17477) to produce lentiviral particles, as previously described[34]. NCI-H23, NCI-H1299 and LLC1 were transduced with lentiviral particles and Polybrene Reagent (Sigma-Aldrich, 5 μg/ml). Puromycin selection started the day after infection for 7 days, using a range between 1–2 μg/ml in complete medium. Luciferase expression was confirmed by Dual-Glo® Luciferase assay (Promega) and Glomax® Discover Microplate Reader (Promega).

## In vivo mouse experiments

All animal experiments were carried out in strict accordance with Italian laws (D.L.vo 26/2014 and following additions) and approved by institutional and national committees (Authorization 780/2020-PR released by the Italian Ministry of Health). Mice were housed under pathogen-free conditions in the animal facilities at the European Institute of Oncology–Italian Foundation for Cancer Research (FIRC) Institute of Molecular Oncology (IEO–IFOM, Milan, Italy). Studies were performed on 6–8 week-old nonobese diabetic severe combined immunodeficient (NOD/SCID) interleukin-2 receptor γ (IL-2Rg)-null (NSG) male mice or on 6 week-old C57-BL/6 immunocompetent female mice. Mice were housed under pathogen-free conditions in the animal facilities at the European Institute of Oncology–Italian Foundation for Cancer Research (FIRC) Institute of Molecular Oncology (IEO–IFOM, Milan, Italy). A maximum number of 5 mice were housed in individually ventilated cages at a temperature of 23 °C, 40% humidity, with a circadian cycle of 12 h light/12 h dark. Water and food were provided *ad libitum*. Mice were euthanized through $CO_2$ inhalation and tumor tissues were collected for laboratory analyses. Samples were kept refrigerated before processing. NSG mice were purchased from Charles River Laboratories (Lecco, Italy), whereas CD57-BL/6 were obtained from Envigo (Indianapolis, IN, USA).

For xenograft engraftment, NSG mice were injected with $2 \times 10^6$ NCI-H23-LUC cells (subcutaneously) or with $0.3 \times 10^6$ NCI-H1299-LUC cells (intravenously) (Fig. 6A). For the syngeneic model, C57-BL/6 mice were injected with $0.5 \times 10^6$ LLC1-LUC cells (subcutaneously) (Fig. 7A). Tumor growth was weekly monitored using In Vivo Imaging System (IVIS, PerkinElmer, Waltham, MA, USA). Briefly, mice were intraperitoneally injected with 150 mg/kg of XenoLight D-Luciferin–Potassium Salt Bioluminescent Substrate (PerkinElmer). After 10 min, animals were anaesthetized with an isoflurane apparatus. Luminescence images were acquired using Living Image Software v4.0 (PerkinElmer). When a basal bioluminescence signal was detected (6–13 days after tumor injection according to distinct mouse models), mice were randomized into different experimental groups ($n = 5$–8 each arm). Mice that did not show engraftment were excluded. The maximal permitted tumor size corresponded to a tumor bioluminescence up to $10^9$ photons/second. Mice that reached this endpoint were immediately euthanized before the end of the experimental procedures. The maximum permitted tumour burden was not exceeded.

Xenograft studies displayed 4 experimental arms: vehicle-treated group, OTX015-treated group, NK92-treated group, and OTX015 + NK92-treated group. OTX-015 was dissolved in a vehicle (10% DMSO, 40% PEG300, 5% Tween80, 45% saline) and intraperitoneally administered 3 times a week (#HY-15743, MedChem Express, at the concentration of 5 mg/Kg). $0.5 \times 10^6$ NK92 cells per mouse were intravenously injected once a week for 3 weeks together with Recombinant Human IL-2 (Peprotech, 2000 IU/mouse). For the subcutaneous growth of NCI-H23 cells, tumor size was assessed by digital calliper and by bioluminescence imaging signal using ImageJ software, by converting pixel intensity into centimetres and determining tumor mass sphere volume in $mm^3$. For the orthotopic growth of NCI-H1299 cells, radiant efficiency was calculated based on the epifluorescence signal, as indicated in the user manual of the IVIS system (PerkinElmer). Mice were euthanized 27–33 days after tumor injection through $CO_2$ inhalation and tumor tissues were collected for further analyses.

Syngeneic studies displayed 4 experimental arms: vehicle-treated group, OTX015-treated group, NK neutralizing monoclonal antibody (αCD122)-treated group, and OTX015 + αCD122-treated group. NK neutralizing anti-mouse CD122 monoclonal antibody (#BE0298, BioXcell, Lebanon, NH, USA) was intravenously administered twice a week at the concentration of 300 μg/mouse for a total of five doses, as previously described[70]. OTX-015 was dissolved in a vehicle (10% DMSO, 40% PEG300, 5% Tween80, 45% saline) and intraperitoneally administered 3 times a week at the concentration of 5 mg/Kg, as the xenograft models described above. The efficacy of NK depletion was checked by flow cytometry assessment of tumor specimens. Tumor growth was assessed using both digital caliper and bioluminescence imaging signal through ImageJ software, by converting pixel intensity into centimeters and determining tumor mass sphere volume in $mm^3$. Mice were euthanized 21 days after tumor injection through $CO_2$ inhalation and tumor tissues were collected for subsequent analyses.

## Mouse tissue processing

Tumors and lungs collected from mice were kept in a storage medium (RPMI medium added with antibiotics) at 4 °C and processed within 24 h from mouse sacrifice. Tissues were enzymatically digested with Liberase™ DH Research grade (Sigma-Aldrich, 0.26 IU/ml) for 1 h at 37 °C. The cell suspension was filtered through a 40 μm strainer to remove tissue debris. Red Blood Cell Lysis Buffer (Sigma-Aldrich) was used to remove erythrocytes. Cells were resuspended in FACs Buffer and NK92 infiltration was detected by flow cytometry analysis with BD FACS Canto II.

## Immunohistochemistry

Mouse tumor specimens were formalin-fixed and paraffin-embedded (FFPE). Immunohistochemistry was performed on representative 4 μm FFPE consecutive sections. Tissue sections were retrieved with EDTA buffer (pH 8) at 95° for 72 min. Slides were stained with rabbit monoclonal anti-human CD56 antibody (MRQ-42, Cell Marque, Sigma-Aldrich, 1:100) or mouse monoclonal anti-human CD45 antibody (2B11, PD7/26, Cell Marque, Sigma-Aldrich, 1:100) and developed on Ventana Bench-Mark ULTRA platform (Ventana-Roche, Tucson, USA) using the UltraView DAB detection Kit (Ventana-Roche). Slides were counterstained with haematoxylin.

## Statistical analysis

Statistical analyses were performed using GraphPad Prism Software v9.3.0 (GraphPad Software). Statistical significance was determined using two-tailed Student's *t*-test. Each experiment was replicated multiple times (>3 up to 6). Each Western Blot analysis was repeated with similar results in at least two independent experiments. If not otherwise specified, values are expressed as mean ± SEM (Standard error of the mean). Adjusted *p*-value was calculated by correcting the *p*-value for multiple testing using Benjamini-Hochberg's method.

The statistical analysis assessing in vivo antitumor activity relies on the ratio between the treatment and vehicle-treated mice area under the curve (AUC) corresponding to the tumor growth divided by the observation time[71]. Assuming a coefficient of variation (CV) of AUC equal for each arm, at least 5 mice per arm were necessary to detect, with 0.8 potency, a halving of tumor growth (alpha error equal to 0.05). Statistical power was calculated by applying G*Power software v3.1.9.7 (http://www.gpower.hhu.de). Two-way ANOVA with multiple comparison (Tukey test) was applied to detect significant differences among experimental arms.

## Reporting summary

Further information on research design is available in the Nature Portfolio Reporting Summary linked to this article.

## Data availability

The RNA-seq data generated in this study have been deposited in the ArrayExpress database under the accession code E-MTAB-12853 and in the ENA database under the accession number ERP146384 (https://www.ebi.ac.uk/ena/browser/view/PRJEB61285). The following ChIP-seq public datasets were analysed: GSE156423 (https://www.ncbi.nlm.nih.gov/geo/query/acc.cgi?acc=GSM4730573) ChIP-seq for BRD4 in NKs isolated from peripheral blood of two healthy donors[29], GSE101225 (ENCODE project n°ENCSR583ACG; https://www.encodeproject.org/experiments/ENCSR583ACG/) ChIP-seq for BRD4 in K562 cell line, GSE231137 (ENCODE project n°ENCSR140GLO; https://www.encodeproject.org/experiments/ENCSR140GLO/) ChIP-seq for SMAD3 in K562 cell line[30]. The remaining data are available within the Article, Supplementary Information or Source Data file. Source data are provided with this paper.

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

## Acknowledgements

We wish to thank Marina Grassi and Cristian Ascione for their technical support and all the members of the lab for the stimulating discussion. This work was funded by the Italian Ministry of Health as part of the program "5perMille, year 2016" promoted by the AUSL-IRCCS of Reggio Emilia to VS. This study was partially supported by Italian Ministry of Health—Ricerca Corrente Annual Program 2025 and by AIRC (Associazione Italiana Ricerca contro il Cancro) IG21772 grant to AC, IG26377 grant to VS and IG20109 grant to FB. FR was supported by Fondazione Umberto Veronesi.

## Author contributions

F.R. performed experiments, data collection and sample analyses, data interpretation, manuscript writing and revision. G.G., F.T., V.M., E.S. contributed to data collection, analysis and interpretation. E.Z., S.P., F.L., M.P. contributed to patient specimen collection and clinical data analyses. G.T., P.F., S.O. executed mouse experiment and data collection. A.C., F.B. contributed to study supervision and manuscript revision. V.S. contributed to study supervision, data interpretation, manuscript writing and revision. All authors approved the final version of the manuscript.

## Competing interests

The authors declare no competing interests.

## Inclusion and ethics

The research workflow was discussed amongst all collaborators and experimental work was executed accordingly. Researchers were involved at all locations in Italy where the research work was conducted. The experiments on humans and on mice were conducted according to Italian laws and institutional guidelines and approved by the competent Committees: animal experiments were approved by institutional and national committees (Authorization 780/2020-PR released by the Italian Ministry of Health). The study on humans was approved by the local ethical committee (Comitato Etico dell'Area Vasta Emilia Nord (AVEN)—Reggio Emilia district, Italy, authorization code 196/2017). All patients were informed and signed a written consensus. Patients had the right to withdraw their consensus at any time. Biological and chemical risk for researchers was managed according to Institutional guidelines.
