## [Peer Review File · Nature Communications]

Reviewers' Comments:

Reviewer #1:

Remarks to the Author:

The manuscript identifies novel molecular targets in NK cell effector function, whose inhibition enhanced NK cytotoxicity.

The proof-of-concept is that administration of the epigenetic drugs BETi can significantly increase the efficacy of adoptive NK cell therapies in NSCLC mouse models and ex-vivo patient derived 3D models.

The data are clear. I have some concerns on the relevance of the models used to the human condition.

Specifically, the authors are using alloreactive NK cells; either NK cell lines and third party NK cells and measuring reactivity against allo-NSCLC lines from the NCI60 bank. This mismatch biology is polar opposite of patients endogenous NK cell reactivity to self tumor tissue. If the goal is to establish a potential IO treatment for NSCLC then this BETi approach should be better examined in synergic models of NSCLC. Further, if the goal is to prove this approach may be useful in allo-NK cell adoptive therapy approaches then the authors would be best served using models that reflect clinical evaluation of allo-cell therapies, namely DCBL or MM. NK cell therapy for NSCLC is not in the minds of most clinicians due to the absence of impactful data for cell therapies in solid tumors.

Simialrly, the spheroid work is all allo-reactive NK cells thus again, hard to imagine how the BETi would payout when self HLA-C, HLA-E, HLA-F are expressed on the target cells

Reviewer #2:

Remarks to the Author:

Reggiani and colleagues show that the BET inhibitor OTX015 reduced the expression of inhibitory receptors on NK-92 cells via cooperation between BRD4 and SMAD3. Treatment with OTX015 increase in vitro and in vivo activity of NK-92 cells. The manuscript present interesting data and the conclusions are for the most part supported by the results. However, the authors need to address certain aspects to improve the quality of the manuscript.

It is unclear regarding experimental setup in certain cases. For instance, were cells pre-activated with cytokines in figure 4B? In co-cultures with tumor cells (e.g. figures 1 and 5), were NK cells treated with BETi or were also tumor cells exposed to BETi? This is important to understand since JQ1 has been shown to reduce the expression of not only PD-1, but also PD-L1. Also, a figure describing the treatment schedule for the in vivo experiments would be helpful to the readers.

Throughout the manuscript, OTX015 was used and only selected experiments were performed with JQ1. Since the authors claim that BET inhibitors drive NK cell activation, selected experiments should be included using the JQ1 drug.

To increase the translational impact of the manuscript, functional assays with primary NK cells silenced for SMAD3 should be performed. For example, selected experiments shown in figures 5 should be re-produced with primary NK cells.

To confirm the transcriptomic data, protein expression analysis (e.g. flow cytometry or WB) should be performed for selected immune checkpoint receptors on NK-92 cells and primary NK cells.

In figure 1, there is a dramatic decrease in the frequency of IFNg+ CD4 T cells in SQ lung cancer, but not in AD lung cancer. This finding should be discussed and additional SQ samples should be included to confirm this reduced frequency.

Reviewer #3:

Remarks to the Author:

The manuscript "BET inhibitors drive Natural Killer activation in non-small cell lung cancer by striking BRD4 and SMAD3 cooperation" by Reggiani et al., described work investigating the capacity of BETi to stimulate an anti-tumoral immune response against NSCLC, specifically, activating NK cells. The research also aimed to dissect the molecular mechanism underlying the impact on immune cell function.

First, it was claimed that BETi stimulated interferon production from tumor-derived immune cells. Next, it was shown that BETi works in synergy with NK cells to elicit a pro-apoptotic response against tumor-derived spheroids. This killing appears to be dependent upon SMAD3-driven transcription. The final part of the study looks at NK cell + BETi cooperativity against xenografts using an immunocompromised NSG model.

Overall, the concept that BETi co-operates with NK cells to repress tumor growth is quite novel and a potentially important finding that adds considerably to the field and may likely have clinical implications. A shortcoming of the study is that the work supporting a focus on NK is not convincing, and it is unclear that these cells potentiate anti-tumor responses more so than other lymphocytic populations. The study would also be strengthened by a thorough ChIP-seq analysis of publicly available datasets. The in vivo data are quite impressive, but suffers from the use of an immunocompromised model.

Points for revision:

Figure 1:

1a. The first experiment described within the text examined the impact of very high quantities of BETi against immune cells isolated from primary lung tumors.

- There is no justification for the use of 1 micromolar BETi. In vitro, these drugs, including OTX015, have been shown to have anti-proliferative properties at nanomolar ranges. It is challenging to believe that off-target effects are not being observed at this concentration. Certainly, in clinical trials, this concentration could not be achieved within the plasma.

A dose response, showing the concentrations required for target induction, such as HEXIM1, MYC repression, or the targets identified in figure 3 by RNA-seq.

1b. The stimulation of interferon production in Fig 1a is marginal at best, especially when considering the large amount of drug being employed. How does this compare to other activators of immune cell response such as PMA alone? For this experiments +/- PMA could be shown.

1c. For the co-culture experiments described in figure 1c, the addition of JQ1 alone significantly impaired cell proliferation. Why is this happening? How was this experiment done? Could JQ1 diffuse across a barrier to the target cells? How was this experiment different from adding JQ1 directly onto the target cells?

1d. For figure 1d, why wasn't Annexin V measured in response to BETi alone?

1e. It is stated that "NK cells purified from the TIL fraction were sufficient alone to obtain the highest anti-tumor activity, further supporting the central role of NK cells in triggering anti-tumor response and the specific effect of BETi in enhancing NK cytotoxicity compared to other TME immune populations".

- Where is the comparison with Annexin V levels in other TIL populations? I cannot find this. Within the text, stating the amount of Annexin V positive cells when exposed to BETi+NK versus BETi+CD8 or BETi+CD4 populations would be helpful.

1e. Overall, the concept that BETi promotes anti-tumor cytokine responses primarily through NK cells could be strengthened.

2. Figure 2.

2a. It is claimed that "an intermediate dosage (1 μ M) of the drugs that had a limited effect on tumor spheroid integrity" was chosen.

At what concentration does OTX015 prohibit BRD4 binding in these cells? 1 micromolar is a high concentration of drug. Again, sensitive cells respond to nanomolar doses of OTX015. At what concentration of OTX015 could transcriptional responses be observed (eg HEXIM1 upregulation, MYC down-regulation)?

2b. For figure 2, the focus on NK cells is not strongly justified by the data in figure 1. Would the same response be observed using T cells or B cells?

3. Figure 3

3a. The time of OTX015 treatment for the RNA-seq experiment, and concentration of drug employed, should be stated within the main text, and within the figure legend. Currently, I cannot find this information anywhere within the manuscript. Thus, the robustness of these RNA-seq data are questioned.

3b. For figure 3H, it seems unusual that BRD4 can be found at the TSS of all of these genes. Can these data be corroborated by publicly available ChIP-seq datasets? Also, can the authors prove specificity of the anti-body? Are there genes TSS where BRD4 is not found?

3c. There are many ChIP-seq datasets available for both SMAD3 and BRD4. It is essential that these datasets be computationally analyzed for overlap between BRD4 and SMAD2/3 binding at promoter and enhancers. Currently, the ChIP data are not convincing and lack negative controls.

Figure 4:

4a. Was SMAD3 identified as a OTX target within the RNA-seq data? What about other SMADs?

4f. Can this data be corroborated by ChIP-seq datasets for BRD4?

Figure 5:

5a. A brief description of "immune synapse" and the relevance might be provided within the text.

Figure 6:

6a. The choice of using an immunocompromised mouse model when studying immune modulation is unusual. The data would be much more relevant if an immunocompetent model were employed.

6b. The concentration of OTX015 used in vivo should be stated within the figure legend. It appears to be 5 mg/Kg which is a very reasonable concentration, lower than that employed by most studies.

6c. The number of mice used for for each cohort should be either shown in the legend of Figure 6a or stated within the figure legend.

6d. For the xenograft experiments, how long after tumor implantation were drug treatments initiated?

6e. Can it be shown in the xenograft model employed, that NK cells introduced to the mice accumulate at the tumor site? As it stands, it is possible that that effect of NK cells on tumor growth is indirect.

Reviewer #4:

Remarks to the Author:

Manuscript by Reggiani et al "BET inhibitors drive Natural Killer activation in non-small cell lung

cancer by striking BRD4 and SMAD3 cooperation",

Reggiani et al investigate the potential utility of BET inhibition as an immunomodulatory agent. Using a combination of patient samples, mouse models and cell lines, the authors evaluate the effects of epigenetic drugs on NK cell cytotoxic potential. The authors posit that a BRD4-SMAD3 axis regulates an inhibitory program, including checkpoint receptor expression in NK cells, and that BET inhibition results in an epigenetic reprogramming leading to increased NK cell cytotoxicity and tumor control. Overall, the topic is of interest, as the therapeutic approach leverages epigenetic regulation of immune response as an anti-tumor approach. However, several concerns need to be addressed to strengthen the interpretation and conclusion of the data presented in the current form.

Major comments:

- BET inhibitors alone seem to have a large effect on the NSCLC lines (Fig. 1C-1F, Fig. 2B, Fig 6A-6I). The authors focus on the effects of combination of NK cells with BET inhibitors for the majority of their experiments, and on the effects of the drugs on the NK cells themselves. However, a more detailed characterization of the effects of BET inhibitors on NSCLC cells alone should be performed to delineate the cancer-cell autonomous effects of BET inhibition to allow for a more robust conclusion on additive effects vs synergistic effects of drug and NK treatment.
- In Fig. 1A-1B, the authors identify NK cells as a population of interest due to the enhancement of IFN γ production in these cells by treatment with BET inhibitors. However, in Fig. 1I, there is no difference in IFN γ production but rather a difference in CD107a expression. Can the authors provide some explanation for the discrepancy here? Are the degranulating NK cells using other cytokines or cytotoxic effector proteins? IFN γ production is again very low in Fig. 2E-2F.
- In Fig. 3, the authors identify that BETi treatment results in significant differential expression of >3000 genes. The authors subsequently focus on the many inhibitory surface markers, describing this as an NK exhaustion signature. They validate the differential expression of these surface markers in a replicate experiment (Fig. 3D) and in NSCLC purified NK cells (Fig. 3E), which is critical, and confirm that BRD4 is the critical component by knockdown and pulldown experiments. However, the authors do not further explore the functional relevance of these surface markers to the phenotypes that they observe. A number of these markers have specific therapeutic antibodies which could be used to test how much they contribute to the BETi phenotypes in vitro and ex vivo.
- The authors indicate that siSMAD3 enhances the anti-tumor efficacy of NK cells (Fig. 5A-5B), and that it is an incomplete phenocopy of BET inhibition. Overexpression or constitutive activation of SMAD3 would help to position it downstream of BRD4, and more precise examination of how much the SMAD3 activity impacts the BET inhibition phenotype.
- In Fig. 6A-6D, the authors grow tumors in NSG mice, then subsequently treat with or without an NK cell line and with or without OTX-015. The authors mention that the tumors were palpable when treatment was initiated. Looking at Fig. 6A, tumors in the combination group look undetectable for the majority of the experiment, and the groups are randomized before there is an ability to measure tumor size, and hence most any discernible tumor burden. While these data show convincingly that the combination treatment allows for large scale tumor control in this context, the efficacy of these treatments would be better evaluated in an experimental schema where there was an established tumor which was then challenged with the treatment. The statistics in panel A also are compared only to Vehicle control. It seems likely that with multiple comparisons corrections there would not have been an observed difference between NK-treated and NK+OTX-treated groups, judging based on the error bars and the length of the bottom left y-axis. Similar principles apply to the tail vein model experiments, but I acknowledge that there does seem to be better engraftment in the lungs than outgrowth in the flank.
- Fig. 6K would benefit from the phenotypic evaluation of NK cells isolated directly ex vivo from tumors to see how they interact with tumors in situ, with both surface markers for activation and exhaustion status and functional molecules like cytokines and effector proteins.

Minor comments:

- Are there differences in phenotypes between CD56 bright and CD56 dim induced by BETi?
- Some references didn't load in right- numbers 22 and 29, for example.

We wish to thank the Editor and the Reviewers for the opportunity to present this revised version of our manuscript in which we attempt to address all the concerns and issues that were raised during the revision process.

First, to increase the impact of our finding and to further support our conclusions, we performed an additional set of *in vivo* experiments, including a new syngenic model of NSCLC. The results of this experiments have been included as the newly added Figure 7.

Moreover, we processed and analysed 8 additional fresh patient-derived NSCLC surgical specimens to answer to the Reviewers' requests to provide more data and experiments, including the analyses of different TIL populations (namely CD8+ T cells and CD4+ T cells) and the successful knock-down of SMAD3 in patient-derived intratumor NKs.

In parallel, we performed bioinformatic analyses using publicly available ChIP-seq datasets and corroborated our proposed molecular mechanism of BRD4 and SMAD3 cooperation and transcriptional regulation.

Below, you can find a point-to-point answers to the Reviewers.

We believe that this revised version of manuscript has been significantly improved and we wish to thank the Reviewers for their useful and constructive comments.

REVIEWER COMMENTS

Reviewer #1 - NK cell therapy, epigenetics (Remarks to the Author):

The manuscript identifies novel molecular targets in NK cell effector function, whose inhibition enhanced NK cytotoxicity.

The proof-of-concept is that administration of the epigenetic drugs BETi can significantly increase the efficacy of adoptive NK cell therapies in NSCLC mouse models and ex-vivo patient derived 3D models.

The data are clear. I have some concerns on the relevance of the models used to the human condition.

Specifically, the authors are using alloreactive NK cells; either NK cell lines and third party NK cells and measuring reactivity against allo-NSCLC lines from the NCI60 bank. This mismatch biology is polar opposite of patients endogenous NK cell reactivity to self tumor tissue. If the goal is to establish a potential IO treatment for NCSLC then this BETi approach should be better examined in synergic models of NSCLC.

We thank the Reviewer for the comments. To address this point, we performed novel *in vivo* experiments using a syngenic mouse model that was included in the revised Fig.7.

We used the Lewis Lung carcinoma (LLC1) syngenic model obtained in the C57-BL/6 background. To validate the essential role of NKs in mediating the BET inhibitor anti-tumor efficacy, we treated

mice with the BET inhibitor OTX015 in presence or not of the NK-neutralizing antibody (α mouse CD122) that induced NK cell depletion.

In this context, we found that NK presence was essential in mediating the anti-tumor immune response, and that the NK depletion was sufficient to impair anti-tumor immune surveillance boosting tumor growth (Revised Fig.7B-E). Moreover, OTX015 efficacy was found significantly impaired when NKs were depleted, supporting our conclusions that NKs are fundamental for the anti-tumor activity of BET inhibitors. We also analysed the mouse immune infiltrate in tumor tissues and we demonstrated that the activation induced by OTX015 was highly specific for NKs that displayed a significant increase of the main degranulation marker CD107a and a concomitant reduction of Immune-checkpoint (IC) expression, including PD1, CTLA4, TIGIT (Revised Fig.7F-L).

Further, if the goal is to prove this approach may be useful in allo-NK cell adoptive therapy approaches then the authors would be best served using models that reflect clinical evaluation of allo-cell therapies, namely DCBL or MM. NK cell therapy for NSCLC is not in the minds of most clinicians due to the absence of impactful data for cell therapies in solid tumors.

The main purpose of our paper was not to validate an adoptive NK cell therapy application for NSCLC patients, even if there are already clinical trial results supporting the relevance of allogenic NK cell administration in this context (PMID: 34332557; 32027620). We believe that this work can be a preliminary proof-of-principle of the potential clinical application of BET inhibitors to potentiate NK effector properties in NSCLC patients. Future studies will be required to properly address this concept and develop specific therapeutic strategies for patients.

Of note, we have strong evidence that BETi ability to enhance NK cell anti-tumor activities is not limited to NSCLC. We believe that adding another tumor model (such as DLBCL or MM) is out of the scope of the present paper. However, similar experiments are currently being carried out in collaboration with our co-authors (Francesco Bertolini's group from IEO Hospital) and the results will be included in a future manuscript. Here, we show, confidentially to the Reviewers, the preliminary results on a Diffuse Large B-Cell Lymphoma (DLBCL) xenograft model. *In vivo* studies were carried out in immunodeficient NOD.Cg-PrkdcSCIDIL2rgtmWjl/SzJ (NSG) mice that were subcutaneously engrafted with 3×10^6 patient-derived primary tumor (LNH1) cells. LN1 were derived from a DLBCL patient and were successfully applied to NSG models to analyse the *in vivo* NK cell cytotoxicity, as previously described (PMID: 300832564).

The results clearly demonstrated that NK92 cells plus BETi administration had a better impact on tumor growth compared to single-agent therapies, as we previously demonstrated in our NSCLC models. Moreover, we analysed the exhausted/activated profiles of NK92 infiltrated in DLBCL tumors collected from these mice by flow cytometry. We confirmed that BETi increased NK activation (CD107a surface expression and IFNG production) and down-regulated the expression of ICs *in vivo*.

Figure legend: We performed a subcutaneous injection of 3×10^6 LNH1 cells in NSG mice ($n=5$ for each arm) and therapies were started 13 days after tumor injection. NK92 cell were intravenously administered once a week (0.5×10^6 cells/mouse), whereas the BET inhibitor OTX015 was administered three times a week (5mg/Kg). Tumor growth was monitored through digital calliper. *** $p < 0.001$

[Redacted]

Figure legend: Flow cytometry analysis of DLBCL xenograft tumors. Tumor-infiltrated NK92 displayed an increased activated profile when OTX015 was concomitantly administered to mice. IFNG production and CD107a surface expression were found increased, indicating an increased NK degranulation and cytotoxicity. Moreover, IC surface expression (PD1, CTLA4/CD152, TIGIT) was detected impaired, confirming the reduction of NK exhaustion triggered by BETi.

Similarly, the spheroid work is all allo-reactive NK cells thus again, hard to imagine how the BETi would play out when self HLA-C, HLA-E, HLA-F are expressed on the target cells

We thank the Reviewer to raise this point. We need to point out that Figure 2 from the original manuscript displayed the use of autologous NK cells and tumor spheroids (CTOS) that were purified from the same patient-derived lung tumor tissue applying the procedure described in the M&M section. This protocol allowed the separation of both immune infiltrate and tumor spheroids from the same specimen. We showed in Figure 2 and revised Suppl.Fig.3N that the anti-tumor potential of autologous NKs were enhanced by BETi as well.

Reviewer #2 - NK cell biology, immunotherapy (Remarks to the Author):

Reggiani and colleagues show that the BET inhibitor OTX015 reduced the expression of inhibitory receptors on NK-92 cells via cooperation between BRD4 and SMAD3. Treatment with OTX015 increase in vitro and in vivo activity of NK-92 cells. The manuscript present interesting data and the conclusions are for the most part supported by the results. However, the authors need to address certain aspects to improve the quality of the manuscript.

We thank the Reviewer for the positive comments on our work. We tried to improve our manuscript by addressing all the points raised by the Editor and the Reviewers. We believe that the manuscript is now much more solid thanks to additional experimental data and analyses that were included in the revised draft.

It is unclear regarding experimental setup in certain cases. For instance, were cells pre-activated with cytokines in figure 4B?

We are sorry that the experimental procedures were not exhaustive. In this revised version of the manuscript, we improved the experimental description in the M&M section and throughout the manuscript to help comprehension of the applied experimental design.

In detail, in figure 4B, NKs were purified from fresh patient tumors and were resuspended in RPMI-medium added with 10%FBS, antibiotics, and IL2. Vehicle (DMSO) or BETi were added directly to the medium of these freshly purified NKs without cytokine pre-activation. After 24h, cells were harvested to extract RNA for subsequent analyses.

In co-cultures with tumor cells (e.g. figures 1 and 5), were NK cells treated with BETi or were also tumor cells exposed to BETi? This is important to understand since JQ1 has been shown to reduce the expression of not only PD-1, but also PD-L1.

In all co-culture experiments, both the ones employing cell lines or primary cells (Figures 1, 2, 5 and Suppl. Fig. 2, Suppl. Fig. 3, Suppl. Fig. 4, Suppl. Fig. 7, Suppl.Fig.8), immune cells were pre-treated with BETi 24h prior of being co-cultured with tumor cells. Then, the co-cultures were maintained under BETi treatment for all the duration of the experiment, thus both immune and tumor cells were exposed to drugs during the co-culture. We modified the M&M section to clarify this point.

As correctly mentioned by the Reviewer, the role of BETi in regulating PD-L1 in tumor cells is well established. To address this issue, we analysed the expression of a panel of IC molecules in three different NSCLC cell lines and patient-derived tumors upon BETi treatment (Suppl.Fig.9A-E). As expected, PDL-1 was severely down-regulated by BETi. Besides, we observed a dramatic down-regulation of other ICs, including PD-L2, LSGAL9, and HVEM. We also observed a concomitant down-regulation of HLA class I molecules on tumor cells that can further trigger NK target recognition and effector functions. We added these data to the revised version of the manuscript (Suppl.Fig.9A-E) and addressed this issue in the Discussion section.

Also, a figure describing the treatment schedule for the *in vivo* experiments would be helpful to the readers.

We added the *in vivo* schedules for mouse experiments, that now are provided in revised Figure 6A and revised Fig.7A.

Throughout the manuscript, OTX015 was used and only selected experiments were performed with JQ1. Since the authors claim that BET inhibitors drive NK cell activation, selected experiments should be included using the JQ1 drug.

As requested, we performed additional experiments using JQ1 and added the relative panels, including Suppl-Fig.2C,2E (co-cultures of purified patient-derived NKs with NSCLC cells, including both proliferation assay and apoptosis assessment), Suppl.Fig.2L (ELISA for IFN- γ), Suppl.Fig.3G-J (co-cultures of CD4/CD8/NKs patient-derived cells with NSCLC cells, including proliferation and apoptosis analyses), Suppl.Fig.4E,J-K (co-cultures of NSCLC tumor spheroids with NK92, including both proliferation and immunological synapse analyses), Suppl.Fig.5D-F (flow cytometry analysis of IC on NK92 and patient-derived NKs), Suppl.Fig.5G (dose response on NK92 cells), Suppl. Fig. 6G,I-J (ChIP analysis for BRD4 and SMAD3), Suppl.Fig.9A-E (expression of IC and MHC-I in several NSCLC cell lines and patient-derived CTOS), Suppl.Fig.9F,9H (dose response on NSCLC cells). JQ1 fully recapitulated the effects that were previously observed with OTX015 treatment.

To increase the translational impact of the manuscript, functional assays with primary NK cells silenced for SMAD3 should be performed. For example, selected experiments shown in figures 5 should be re-produced with primary NK cells.

We wish to thank the Reviewer for this relevant suggestion. As requested, we performed SMAD3 KD experiments in patient-derived NK cells.

Briefly, intratumor NKs were freshly isolated from AD tumors right after surgery (n=3) and transfected with Lipofectamine with siRNAs targeting SMAD3. After checking the efficiency of SMAD3 down-regulation (revised Fig.5G), we confirmed that this determined a significant down-regulation of IC expression also in patient-derived NKs (revised Fig.5G). Moreover, we performed functional experiment, by establishing co-cultures of SMAD3^{KD} NKs with NCI-H23 or NCI-H1299.

The co-cultures were monitored overtime and analyzed through Incucyte platform (revised Fig.5H-I, revised Suppl.Fig.7G). We confirmed that the silencing of SMAD3 was sufficient to improve the ability of primary NKs to recognize target tumor cells, form immunological synapses and contrast tumor cell proliferation.

To confirm the transcriptomic data, protein expression analysis (e.g. flow cytometry or WB) should be performed for selected immune checkpoint receptors on NK-92 cells and primary NK cells.

As requested, we performed flow cytometry analysis of the main immune checkpoint receptors (i.e. PD-1, CTLA4, TIGIT, TIM3, LAG3) on both NK92 and patient-derived intratumor NKs treated with BETi (JQ1 or OTX015) at different time points, namely 48h and 72h (Revised Suppl.Fig.5D-F). The results fully confirmed the transcriptomic data.

In figure 1, there is a dramatic decrease in the frequency of IFN γ + CD4 T cells in SQ lung cancer, but not in AD lung cancer. This finding should be discussed and additional SQ samples should be included to confirm this reduced frequency.

The Reviewer is right, there was a trend toward reduction of IFN- γ production in CD4+ T cells in SQ samples. However, this difference was not statistically significant likely due to the low number of available cases. We extended this analysis adding two newly enrolled SQ patients that underwent surgery during the time of the revision process (n=4 total patients now). The new graph was included in the revised Figure 1B and confirmed that the effect was not statistical significant.

Reviewer #3 - BETi, epigenetics (Remarks to the Author):

The manuscript "BET inhibitors drive Natural Killer activation in non-small cell lung cancer by striking BRD4 and SMAD3 cooperation" by Reggiani et al., described work investigating the capacity of BETi to stimulate an anti-tumoral immune response against NSCLC, specifically, activating NK cells. The research also aimed to dissect the molecular mechanism underlying the impact on immune cell function.

First, it was claimed that BETi stimulated interferon production from tumor-derived immune cells. Next, it was shown that BETi works in synergy with NK cells to elicit a pro-apoptotic response against tumor-derived spheroids. This killing appears to be dependent upon SMAD3-driven transcription. The final part of the study looks at NK cell + BETi cooperativity against xenografts using an immunocompromised NSG model.

Overall, the concept that BETi co-operates with NK cells to repress tumor growth is quite novel and a potentially important finding that adds considerably to the field and may likely have clinical implications. A shortcoming of the study is that the work supporting a focus on NK is not convincing, and it is unclear that these cells potentiate anti-tumor responses more so than other lymphocytic populations. The study would also be strengthened by a thorough ChIP-seq analysis of publicly available datasets. The in vivo data are quite impressive, but suffers from the use of an immunocompromised model.

We thank the Reviewer for the comments on our work and for raising important point of discussion and improvement of our data.

We addressed all the points raised by the Reviewer to strengthen the focus on NK cells as determinants of BETi anti-tumor immune response in NSCLC. Moreover, we added analysis of available ChIP-seq datasets to corroborate our results obtained in NK92 cells. Finally, we performed additional experiments using a syngenic mouse model with a functional immune system, that confirms our previous data and further supports the determinant role of NKs, compared to other lymphocyte populations, in mediating BET inhibitors efficacy.

Points for revision:

Figure 1:

1a. The first experiment described within the text examined the impact of very high quantities of BETi against immune cells isolated from primary lung tumors.

- There is no justification for the use of 1 micromolar BETi. In vitro, these drugs, including OTX015, have been shown to have anti-proliferative properties at nanomolar ranges. It is challenging to believe that off-target effects are not being observed at this concentration. Certainly, in clinical trials, this concentration could not be achieved within the plasma. A dose response, showing the concentrations required for target induction, such as HEXIM1, MYC repression, or the targets identified in figure 3 by RNA-seq.

We thank the Reviewer to raise this point. The 1 μ M concentration of BETi was chosen based on our previous work on BETi that we published in the context of NSCLC (Gobbi et al., *Oncogene* 2019) and corresponds to the IC50 of NSCLC cells after 72h of incubation. We apologize for this lack of information. We specified that in the M&M section.

We confirmed that this BETi concentration did not affect NK cell viability (as it was shown in the original Suppl.Fig.1C), but was sufficient to stimulate their cytotoxicity and activation toward tumor cells. This concentration is in line with other studies carried out in NSCLC (doi: 10.1172/JCI133090, doi: 10.18632/oncotarget.13181, doi: 10.3390/ijms22041877).

To assess if a lower concentration of BETi can be sufficient to down-regulate BRD4 targets, we performed a dose-response analysis as suggested by the Reviewer. We applied a range from 0.01 to 5 μ M of BETi (both JQ1 and OTX015) and analysed the BRD4 target reduction in NK92 cells, as shown in the revised Suppl. Fig.5G-H. The results confirmed that the 0.5-1 μ M dosage are necessary to get at least 50-70% of BRD4 target reduction after 24h.

Of note, we also analysed SMAD3 expression in these experiments, thus confirming that SMAD3 inhibition is a very specific events that occurs not only in an early-phase of drug exposure, but also at a lower concentration (starting from 0.05 μ M) compared to other established BRD4 targets.

We included these data in our results.

1b. The stimulation of interferon production in Fig 1a is marginal at best, especially when considering the large amount of drug being employed. How does this compare to other

activators of immune cell response such as PMA alone ? For this experiments +/- PMA could be shown.

The effect of PMA/Ionomycin alone on immune cells was already present in the original version of the manuscript as Suppl.Fig.1D.

In this revised version, we also showed flow cytometry analysis of BETi-treated immune cells with or without PMA/Ionomycin stimulation (revised Suppl.Fig.1D).

No IFN- γ + cells was detected in unstimulated conditions, in presence or not of BETi. Conversely, the standard protocol of immune cell stimulation using PMA/Ionomycin for 6h induced a rapid accumulation of IFN- γ in immune cells, as previously described (PMID: 23985769; PMID: 35068054; PMID: 19777549; PMID: 24464647).

The pre-treatment with BETi, followed by PMA/Ionomycin stimulation, enhanced significantly the percentages of IFN- γ + NK cells (up to 80-90%, almost the double amount of positive cells compared to vehicle-treated cells) (Fig.1A-B). Conversely, the percentages of other analysed IFN- γ producing immune populations were not significantly affected by BETi pre-treatment and PMA/Ionomycin stimulation, with only a trend to increase for CD8+ cells (Fig.1A-B).

1c. For the co-culture experiments described in figure 1c, the addition of JQ1 alone significantly impaired cell proliferation. Why is this happening? How was this experiment done? Could JQ1 diffuse across a barrier to the target cells? How was this experiment different from adding JQ1 directly onto the target cells?

The anti-proliferative effect of BETi is well established in many tumor contexts, including NSCLC (PMID: 28838216, PMID: 33760217, PMID: 27607580, PMID: 36448366 and also our previous work Gobbi *et al*, *Oncogene* 2019 PMID: 31406246).

In the present study, we obtained co-cultures putting in “direct contact” the different cell types. We did not apply barriers such as a transwell systems.

The freshly isolated cells (TILs or NKs) were pre-treated for 24h with BETi drugs, then they were red-stained with a viable dye before co-cultures with tumor cells. This was done to monitor immune cells during the real-time imaging analysis. JQ1 or OTX015 were added again directly to the medium of co-cultures or single cultures, thus acting on both tumor and NK cells during the experimental procedure.

We apologize if the experimental details were not clear for this part. We modified the manuscript providing additional detail in our Material and Methods section to clarify this point.

1d. For figure 1d, why wasn't Annexin V measured in response to BETi alone?

The effect of BETi alone in terms of Annexin V levels on tumor cells was already included in the original Suppl.Fig.2 (panels F-H). BETi alone did not affect apoptosis in two NSCLC cell lines (NCI-H23, NCI-H1299) or slightly enhance apoptosis at a very low level in NCI-H1975 cells.

1e. It is stated that “NK cells purified from the TIL fraction were sufficient alone to obtain the highest anti-tumor activity, further supporting the central role of NK cells in triggering anti-

tumor response and the specific effect of BETi in enhancing NK cytotoxicity compared to other TME immune populations”.

- Where is the comparison with Annexin V levels in other TIL populations ? I cannot find this. Within the text, stating the amount of Annexin V positive cells when exposed to BETi+NK versus BETi+CD8 or BETi+CD4 populations would be helpful.

We thank the Reviewer for pointing out this discrepancy on the former version of the manuscript. To avoid overstatements, we performed the requested experiments.

Starting from fresh patient-derived tumor specimens, we purified CD4+ T cells, CD8+ T cells and NK cells. Then, we set up co-culture experiments with NSCLC cells and comparing their anti-tumor activity. We analysed tumor cell proliferation using real-time cell live imaging system (Incucyte) and we evaluated tumor apoptosis with AnnexinV staining by flow cytometry. The results are included in the revised Suppl.Fig.3A-K. As shown by these results, BETi (JQ1 and OTX015) potentiated the cytotoxic activity of CD8+ T cells, but did not have an effect on CD4+ cells. This is in line with other published works that reported the effect of BETi in reducing CD8 T cell exhaustion and in promoting their induction in other tumor contexts and in NSCLC (PMID: 34396987, PMID: 35918330, PMID: 27548527).

Noticeably, BETi enhanced NK cytotoxicity to a higher extent compared to other T cells (Suppl.Fig.3C,3G). Although we cannot rule out the possibility of a wider immune-regulatory activity of BETi in the concerted immune response regulation, we believe that these data further support the relevance of NKs as major targets of BETi. Besides, their novelty further justified the focus on NKs in our manuscript. We added this point to the Discussion section.

1e. Overall, the concept that BETi promotes anti-tumor cytokine responses primarily through NK cells could be strengthened.

We agree with the Reviewer that we could increase the robustness of our choice to analyse NK cells in the present study. Besides the new data available in the Revised Suppl.Fig.3 in which we compared the co-cultures of NKs with CD4+ or CD8+ T cells, we added relevant new data in the revised version of our manuscript with a novel *in vivo* syngenic mouse model (revised Fig.7). In this model, we demonstrated that BETi anti-tumor effect was largely dependent on NK activity, since NK depletion by a monoclonal antibody was sufficient to impair the drug efficacy (revised Fig.7B-E). Besides, *ex vivo* analysis of the tumor-infiltration in BETi-treated mice further supports the notion that BETi have a higher effect on NK cell activation comparing other T cells, by specifically increasing their degranulation (revised Fig.7I-J).

2. Figure 2.

2a. It is claimed that “an intermediate dosage (1µM) of the drugs that had a limited effect on tumor spheroid integrity” was chosen.

At what concentration does OTX015 prohibit BRD4 binding in these cells? 1 micromolar is a high concentration of drug. Again, sensitive cells respond to nanomolar doses of OTX015. At

what concentration of OTX015 could transcriptional responses be observed (eg HEXIM1 upregulation, MYC down-regulation)?

As we clarified above, we previously showed that NSCLC are more resistant to BET inhibitors compared to other tumors. In this manuscript, we applied the 1 μ M dosage based on the results of our previous study on the same NSCLC cell models (Gobbi et al, Oncogene 2019). Besides, this dosage is in line with the one used in other studies employing these drugs in NSCLC cells (doi: 10.1172/JCI133090, doi: 10.18632/oncotarget.13181, doi: 10.3390/ijms22041877).

To be sure that the dosage was correct in primary spheroid tumor models (CTOS), we also performed a dose-curve of both JQ1 and OTX015 on primary CTOS, as reported in the original Suppl.Fig.3 (now revised Suppl.Fig.3L), to exclude over-dosage in case of three-dimensional culture and primary cells. The chosen dosage did not significantly affect CTOS viability or spheroid integrity after 72h, so the effect that we observed in our co-cultures was due to immune cell activation.

As the Reviewer can easily figure out, we can obtain only a small part of the patient surgical specimen, thus the amount of CTOS that we can derive is limited and not sufficient to perform BRD4 ChIP experiments.

To address the Reviewer's comment, we included in the revised manuscript additional dose-response experiments performed in NSCLC cell lines (NCI-H23 and NCI-H1299) of both JQ1 and OTX015, as we previously did for NK92 cells. We applied a range from 0.01 to 5 μ M of BETi and analysed BRD4 target reduction in NSCLC cells, as shown in the revised Suppl. Fig.9F-I. These results confirmed that the 1 μ M dosage is necessary to get at least 50-70% of BRD4 target reduction after 24h. As we showed in our previous work (Gobbi et al, 2019 Oncogene), c-MYC was not targeted by BRD4 in NCI-H23.

2b. For figure 2, the focus on NK cells is not strongly justified by the data in figure 1. Would the same response be observed using T cells or B cells?

As we already mentioned, in this revised version of the manuscript, we performed additional experiments using *ex-vivo* tumor-derived CD4+ T cells, CD8+ T cells and NK cells. We performed new co-culture experiments between purified T cells or NKs with NSCLC cells in presence or not of BETi (JQ1 and OTX015). We demonstrated the superior anti-tumor activity of BETi-treated NKs compared to other immune subsets. We included these new data on tumor proliferation and apoptosis in the Revised Suppl.Fig.3A-K.

We also included novel analyses on syngenic mouse models and demonstrated that NKs were required to observe the *in vivo* BETi anti-tumor effect (revised Fig.7A-E) and that the increased immune cytotoxicity was specific for NK cells, as compared to CD8 T lymphocytes (revised Fig.7I-J).

Regarding B-cells, we analysed by flow cytometry this cell population within the TIL fraction isolated from patient-derived NSCLC. B-cells did not increase IFN- γ production after PMA/Ionomycin stimulation and/or BETi treatment (see Figures below) and thus, we decided to not include them in our subsequent analyses. B cells were also very low represented in our syngenic NSCLC mouse models (revised Fig.7H).

Collectively, these additional new data support and further justify the relevance of focusing on NK cells.

Vehicle + PMA/Ionomycin

OTX015 + PMA/Ionomycin

Figure legend: Flow cytometry analysis of B-cells (CD45⁺CD19⁺) from patient-derived tumors. TILs were freshly isolated and pre-treated for 24h with BETi (1 μ M OTX015) or vehicle (DMSO) before 6h of PMA/Ionomycin stimulation.

3. Figure 3

3a. The time of OTX015 treatment for the RNA-seq experiment, and concentration of drug employed, should be stated within the main text, and within the figure legend. Currently, I cannot find this information anywhere within the manuscript. Thus, the robustness of these RNA-seq data are questioned.

RNA-seq was performed after 24h of 1 μ M OTX015. This information was stated in the M&M section in the original version of the manuscript. We added this information in the results and in figure legend.

3b. For figure 3H, it seems unusual that BRD4 can be found at the TSS of all of these genes. Can these data be corroborated by publicly available ChIP-seq datasets?

BRD4 is an epigenetic reader that recognizes acetylated lysines that are commonly found on transcriptionally active chromatin regions. Thus, it is expected that BRD4 accumulates at the active promoters and enhancers (PMID: 37591883; PMID: 30466442). Here, we show, confidentially to the Reviewers, our unpublished data on ChIP-seq analysis for BRD4 in thyroid cancer (TPC1 cells). These results clearly demonstrate that BRD4 is preferentially detected at the promoters of target genes (45% of BRD4 genomic distribution). Still, BRD4 genomic occupancy is likely to be very context- and cell-specific due to the plasticity of the epigenetic marks recognized by the factor.

[Redacted]

Figure legend: ChIP-seq analysis using anti-BRD4 antibody in TPC1 cells. BRD4 genomic distribution revealed that it is preferentially located on gene promoters.

We also analysed available ChIP-seq publicly datasets for BRD4 in NKs (GEO accession number GSE156423, doi: 10.3389/fimmu.2021.626255) and K562 cells (GEO accession n° GSE101225, ENCODE n°ENCSR583ACG for BRD4).

In primary NKs, we confirmed a significant accumulation of BRD4 on the promoter regions of identified targets, but not in the genomic regions that we used as negative controls in our ChIP experiments, thus confirming the validity of our approach. The results of BRD4 counts were included in the Revised Suppl.Fig.6A.

In the K562 dataset, we observed that the BRD4 transcriptional program was only partially overlapping with the one found in NKs, therefore supporting a context-specificity of BRD4 transcriptional functions. These results were included in the Revised Suppl.Fig.6B.

Also, can the authors prove specificity of the anti-body? Are there genes TSS where BRD4 is not found?

ChIP experiments were performed using the same anti-BRD4 antibody that was used within the ENCODE project in different cell models, including K562 (ENCAB782ZLNQ).

To further prove the specificity of our analysis, we provided additional positive and negative control regions for our ChIP analysis for BRD4, further supporting the specificity of the antibody (revised Suppl.Fig.6F). Moreover, the use of BETi (both JQ1 and OTX015) detached BRD4 from its targets in our ChIP experiments, further confirming the specificity of BRD4 binding to these chromatin regions (Fig.4G,4I and revised Suppl.Fig.6G,I).

Figure legend (now included as revised Suppl.Fig.6F): ChIP analysis using anti-BRD4 antibody in NK92 cells. Negative (-) and positive (+) controls are highlighted. ** p<0.01, *** p<0.001.

3c. There are many ChIP-seq datasets available for both SMAD3 and BRD4. It is essential that these datasets be computationally analyzed for overlap between BRD4 and SMAD2/3 binding at promoter and enhancers. Currently, the ChIP data are not convincing and lack negative controls.

We have included additional negative controls for ChIP experiments (Suppl.Fig.6F).

Transcriptional cooperation is largely context-dependent. Unfortunately, no public dataset for SMAD3 ChIP-seq experiments in NKs was available.

To address the Reviewer's request, we analyzed ChIP-seq datasets for both BRD4 and SMAD3 generated in the lymphoid K562 tumor cell line and available through the ENCODE project (GEO accession n° GSE231137, Encode number ENCSR140GLO for SMAD3; GEO accession n° GSE101225, ENCODE n°ENCSR583ACG for BRD4), even if we are aware that the majority of the targets we identified in our analysis are NK specific and this dataset may not be fully informative.

We detected a significant overlap of BRD4 peaks with SMAD3 peaks (approximately 25%) (Revised Suppl.Fig.6C). Moreover, the meta-profile analysis revealed a relevant enrichment of BRD4 signal on SMAD3 peak regions (Suppl.Fig.6D). In line with these results, a positive correlation between BRD4 and SMAD3 enrichment at the TSS of protein-coding genes was observed (Suppl.Fig.6E).

Overall, these data further support our data and confirm that the transcriptional cooperation between BRD4 and SMAD3. This cooperation was not limited to the NK context. We added these points in the Discussion session.

Figure 4:

4a. Was SMAD3 identified as a OTX target within the RNA-seq data? What about other SMADs?

Yes, SMAD3 was identified as a main target of OTX015 treatment in the RNA-seq, as already reported in Fig.3I. We modified the text and figure legend to make this point more clear.

Beside SMAD3, only SMAD4 was affected by OTX015 treatment with a mild FC (log fold change = 0.53).

4f. Can this data be corroborated by ChIP-seq datasets for BRD4?

We confirmed BRD4 occupancy on SMAD3 promoter genomic regions in both K562 and primary NK ChIP-seq datasets, suggesting that SMAD3 regulation mediated by BRD4 is a conserved mechanism that was not strictly limited to the NK context (see Revised Suppl.Fig.6A-B).

As requested, we confirmed the binding of BRD4 on SMAD3 TSS using publicly available datasets within the ENCODE project. We observed BRD4 ChIP-seq peaks on the SMAD3 promoter in K562 cells, but not in HepG2 cells (Revised Suppl.Fig.6H). Suppl.Fig.6H also reported the exact position of the fragments that we analyzed by ChIP in our analysis (identified as p1, p2, p4) to highlight the overlap with the ChIP-seq tracks.

Figure 5:

5a. A brief description of "immune synapse" and the relevance might be provided within the text.

As requested, we added in the Results section the description of immune synapses and their relevance in the context of immune cell response in the Discussion section.

Figure 6:

6a. The choice of using an immunocompromised mouse model when studying immune modulation is unusual. The data would be much more relevant if an immunocompetent model were employed.

We included in the revised manuscript novel *in vivo* experiments using the Lewis Lung carcinoma (LLC1) syngeneic model in the C57-BL/6 background (Revised Figure 7). To validate the essential role of NKs in mediating the BET inhibitor anti-tumor efficacy, we treated mice with the BET inhibitor OTX015 in presence or not of the NK-neutralizing antibody (α mouse CD122) that induced NK cell depletion.

In this context, we found that NK presence was essential in mediating the anti-tumor immune response, and the NK depletion was sufficient to impair anti-tumor immune surveillance boosting tumor growth (revised Fig.7B-E). Moreover, OTX015 efficacy was found significantly impaired when NKs were depleted, supporting our conclusions that NKs are fundamental for the anti-tumor activity of BET inhibitors. We also analysed the mouse immune infiltrate in tumor tissues and we demonstrated that the activation induced by OTX015 was highly specific for NKs, that displayed a

significant increase of degranulation markers (CD107a) compared to CD8+ T cells (revised Fig.7I-J) and a reduction of Immune-checkpoint (IC) expression (PD1, CTLA4, TIGIT) (revised Fig.7K-L).

6b. The concentration of OTX015 used in vivo should be stated within the figure legend. It appears to be 5 mg/Kg which is a very reasonable concentration, lower than that employed by most studies.

We apologize for the missing information in the main text. The information was included in the M&M section. The Reviewer was very correct, we used 5 mg/Kg of OTX015 for all *in vivo* experiments. We added the drug concentration in Figure Legends.

6c. The number of mice used for for each cohort should be either shown in the legend of Figure 6a or stated within the figure legend.

The number of mice was stated in the Material and Methods section and included in the legends of Figure 6 and revised Figure 7.

6d. For the xenograft experiments, how long after tumor implantation were drug treatments initiated?

As requested by Reviewer 2, we added a time line in revised Figure 6A to clarify the experimental procedures of our *in vivo* studies. The drugs were started when the tumor luminescence was detectable by IVIS system, thus approximately two weeks after tumor injection for NCI-H23 or one week for NCI-H1299.

6e. Can it be shown in the xenograft model employed, that NK cells introduced to the mice accumulate at the tumor site? As it stands, it is possible that that effect of NK cells on tumor growth is indirect.

These controls were already included in the original version of the manuscript in Figure 6 (now revised panels from K to N). We performed both flow cytometry analysis and IHC evaluation of xenografted tumors of NCI-H23 and NCI-H1299 models, in which we demonstrated the recruitment at tumor site of NK92 cells.

Reviewer #4 - Lung cancer immunotherapy (Remarks to the Author):

Manuscript by Reggiani et al "BET inhibitors drive Natural Killer activation in non-small cell lung cancer by striking BRD4 and SMAD3 cooperation",

Reggiani et al investigate the potential utility of BET inhibition as an immunomodulatory agent. Using a combination of patient samples, mouse models and cell lines, the authors evaluate the effects of epigenetic drugs on NK cell cytotoxic potential. The authors posit that a BRD4-SMAD3 axis regulates an inhibitory program, including checkpoint receptor expression in NK cells, and that BET inhibition results in an epigenetic reprogramming leading to increased NK cell cytotoxicity and tumor control. Overall, the topic is of interest, as the therapeutic approach

leverages epigenetic regulation of immune response as an anti-tumor approach. However, several concerns need to be addressed to strengthen the interpretation and conclusion of the data presented in the current form.

We thank the Reviewer for the positive comments on our work. We tried to improve our manuscript and addressed all the raised concerns. We believe that the manuscript is now much more improved by the additional experimental data and analyses.

Major comments:

- BET inhibitors alone seem to have a large effect on the NSCLC lines (Fig. 1C-1F, Fig. 2B, Fig 6A-6I). The authors focus on the effects of combination of NK cells with BET inhibitors for the majority of their experiments, and on the effects of the drugs on the NK cells themselves. However, a more detailed characterization of the effects of BET inhibitors on NSCLC cells alone should be performed to delineate the cancer-cell autonomous effects of BET inhibition to allow for a more robust conclusion on additive effects vs synergistic effects of drug and NK treatment.

We thank the Reviewer for giving us the opportunity to discuss this relevant point. The effect of BETi on NSCLC has been already investigated in a number of works. We also recently contributed to the field, by characterizing the mechanisms of resistance to BETi in NSCLC (Gobbi et al., *Oncogene* 2019). In that manuscript, we showed that BETi affected the expression of YAP and TAZ in NSCLC and we extensively characterized the effect of BETi in reducing tumor proliferation, clonogenic and migration potential. In this work, we aimed to follow up on these results investigating the effect of these drugs on the immune system. Immunotherapy has radically changed the therapeutic approach and perspective for NSCLC patients, thus we believe this is a relevant field that deserves further investigations.

In the revised version of the manuscript, we provided additional analyses focusing on the effects of BETi in modulating the expression of ICs on tumor cells using three different NSCLC cell lines and patient-derived spheroids (CTOS) (Suppl.Fig.9A-E). We showed that BETi induced a significant down-regulation of ICs in tumor cells, triggering NK activation. Besides IC inhibition, we detected a concomitant down-regulation of HLA class I molecules. This is an important finding since the absence of these molecules is associated with an increased ability of NKs to recognize target tumor cells. Taken together, these data suggested that the drug effect may be synergistic with NKs. Future experiments will be required to validate this hypothesis.

We included and discussed this point in the Discussion section.

- In Fig. 1A-1B, the authors identify NK cells as a population of interest due to the enhancement of IFN γ production in these cells by treatment with BET inhibitors. However, in Fig. 1I, there is no difference in IFN γ production but rather a difference in CD107a expression. Can the authors provide some explanation for the discrepancy here? Are the degranulating NK cells using other cytokines or cytotoxic effector proteins? IFN γ production is again very low in Fig. 2E-2F.

We thank the Reviewer for raising this point. The results in Fig.1A-B were obtained after PMA/Ionomycin stimulation that induced a rapid enhancement of IFN γ production. In order to be assessed by flow cytometry, Golgi Plug (Brefeldin A) was added to immune cells to prevent IFN γ release in cell media, leading to intracellular accumulation.

Conversely, Golgi Plug was not added to our co-cultures with tumor cells because we did not want to impair degranulation and immune cell cytotoxicity. In these co-cultures, NK degranulation was predominantly associated with CD107a surface expression and release of IFN γ in culture media. Thus, it is not surprising that the intracellular levels of IFN γ , that we detected by flow cytometry, were lower compared to the ones obtained with PMA/Ionomycin stimulation.

We confirmed that by adding the IFN γ assessment in co-culture supernatants that was performed by ELISA (see Revised Suppl.Fig.2L). We found an increased amount of secreted IFN γ in presence of BETi compared to vehicle in our co-cultures with primary NKs and NSCLC cells.

- In Fig. 3, the authors identify that BETi treatment results in significant differential expression of >3000 genes. The authors subsequently focus on the many inhibitory surface markers, describing this as an NK exhaustion signature. They validate the differential expression of these surface markers in a replicate experiment (Fig. 3D) and in NSCLC purified NK cells (Fig. 3E), which is critical, and confirm that BRD4 is the critical component by knockdown and pulldown experiments. However, the authors do not further explore the functional relevance of these surface markers to the phenotypes that they observe. A number of these markers have specific therapeutic antibodies which could be used to test how much they contribute to the BETi phenotypes in vitro and ex vivo.

Following the Reviewer requests, we performed additional experiments using different IC inhibitors (Nivolumab/anti-PD1, Ipilimumab/anti-CTLA4 and Vibostolimab/anti-TIGIT) alone or in combination with BETi. We performed co-cultures of NSCLC with NK92 cells or patient-derived intratumor NKs. The results are displayed in the revised Fig.5L and Suppl.Fig.8 and demonstrated that OTX015 had a superior ability in enhancing NK cytotoxicity compared to single-agent IC inhibitors. This can be explained by the fact that BETi exerted a concerted down-regulation of multiple ICs and iKIRs in NKs. According to these observations, the addition of IC inhibitors did not improve BETi anti-tumor effect in our co-cultures, further confirming that IC inhibition was a key mechanism of the observed BETi-mediated regulation of NK activity.

We added this point to the Discussion section.

- The authors indicate that siSMAD3 enhances the anti-tumor efficacy of NK cells (Fig. 5A-5B), and that it is an incomplete phenocopy of BET inhibition. Overexpression or constitutive activation of SMAD3 would help to position it downstream of BRD4, and more precise examination of how much the SMAD3 activity impacts the BET inhibition phenotype.

SMAD3 is activated by nuclear translocation upon proper stimulation. Its overexpression may lead to a non-functional accumulation in the cell cytoplasm, without leading to a real activation of its transcriptional program. Therefore, to address the Reviewer's point, we treated NK92 cells with TGF- β , which is a master stimulator of SMAD3. TGF- β stimulation induced the rapid nuclear translocation of the factor and promoted its transcriptional activity in NKs (Revised Suppl.Fig.7I, Revised Fig.5J).

We observed that, under TGF- β stimulation, the expression of several ICs and iKIRs were significantly stimulated (Revised Fig.5J) and immunological synapses were reduced (Fig.5K, Suppl.Fig.7J). We also noted that TGF- β rescued the BETi-induced down-regulation of many of IC and iKIR molecules, in line with the fact that SMAD3 cooperated with BRD4 in their regulation.

In the revised version of the manuscript, these results were further corroborated by public Chip-seq dataset analyses, in which we confirmed that SMAD3 co-localized with BRD4 in binding regulatory elements across the genome (revised Suppl.Fig.6C-E), and by the inhibition of BRD4 by BETi that drastically displaced SMAD3 from IC/iKIR common targets.

In the revised manuscript, we also included SMAD3^{KD} experiments performed in patient-derived NKs (revised Fig.5G-I, revised Suppl.Fig.7G) that confirmed the down-regulation of IC/iKIR previously observed in NK92 cell line. As we already stated in the original version of the manuscript, SMAD3^{KD} was not able to fully recapitulate BETi effects of NKs and thus we did not expect a complete rescue of the phenotype upon SMAD3 induction. Accordingly, BETi effect on NK anti-tumor properties was not completely rescued by SMAD3 induction upon TGF- β treatment.

Collectively, we believe that SMAD3 inhibition is a central part of the BETi-induced activated phenotype in NKs and that the regulation of SMAD3/BRD4 transcriptional complex formation can explain the observed effects on target gene expression. We added these points in the Discussion section of the manuscript.

- In Fig. 6A-6D, the authors grow tumors in NSG mice, then subsequently treat with or without an NK cell line and with or without OTX-015. The authors mention that the tumors were palpable when treatment was initiated. Looking at Fig. 6A, tumors in the combination group look undetectable for the majority of the experiment, and the groups are randomized before there is an ability to measure tumor size, and hence most any discernible tumor burden. While these data show convincingly that the combination treatment allows for large scale tumor control in this context, the efficacy of these treatments would be better evaluated in an experimental schema where there was an established tumor which was then challenged with the treatment. The statistics in panel A also are compared only to Vehicle control. It seems likely that with multiple comparisons corrections there would not have been an observed difference between NK-treated and NK+OTX-treated groups, judging based on the error bars and the length of the bottom left y-axis. Similar principles apply to the tail vein model experiments, but I acknowledge that there does seem to be better engraftment in the lungs than outgrowth in the flank.

As requested by Reviewer 2, we added a time line in revised Figure 6A to clarify the experimental procedures of our *in vivo* studies. The drugs were started when tumor luminescence was already detectable by IVIS system, thus approximately two weeks after NCI-H23 injection or one week for NCI-H1299. Following IVIS evaluation of tumor engraftment, we randomized the mice in the different experimental groups and excluded mice that did not show engraftment. This information was included in the M&M section of the original draft. To further clarify this point, we added additional information in the figure legend and in the M&M section. We hope that now the experimental procedure is more clear.

Revised Figure 6B illustrates the tumor measurement performed with digital calliper, that is less sensitive compared to the IVIS evaluation. Thus, when we started the treatments, tumors were not measurable by calliper but only slightly palpable by the hand of the operator, but detectable by bioluminescence signal. We are sorry that in the main text the word “palpable” was misleading and thus we corrected this statement to be more accurate.

Finally, we added the multiple comparison corrections among different groups in the revised Fig.6B. Indeed, we did not find a statistical significance between the NK vs NK+OTX015 groups when we measured the tumors by digital calliper. However, the tumor growth of these mice was also measured with luminescent analysis which was able to identify statistical significant differences between the NK vs NK+OTX015 groups as shown in revised Fig.6D-E.

- Fig. 6K would benefit from the phenotypic evaluation of NK cells isolated directly *ex vivo* from tumors to see how they interact with tumors *in situ*, with both surface markers for activation and exhaustion status and functional molecules like cytokines and effector proteins.

We thank the Reviewer for the suggestion. Unfortunately, we were not able to perform these analyses on tumor frozen samples that we collected from these xenograft models. The number of events corresponding to tumor-infiltrated NK92 cells was too low after thawing the samples, thus not permitting a complete analysis of activation/exhaustion markers by flow cytometry.

However, we performed a novel set of *ex vivo* analyses on intratumor NKs collected from fresh tumors from our syngenic NSCLC models. The results were included in the revised Fig.7. Flow cytometry analysis indicated that OTX015 significantly increased NK activation (CD107a surface expression, revised Fig.7I-J) and down-regulated several IC molecules, impairing NK exhaustion (PD-1, TIGIT, CTLA3, revised Fig. 7K-L). These results corroborated our *in vitro* and *ex vivo* observations.

Moreover, these data were further supported by the additional results that we showed confidentially in the answer to Reviewer1 on a DLBCL xenograft model in which we assessed by flow cytometry the activation/exhaustion phenotype of tumor recruited NK92 cells. Of note, this model had the same therapy schedule (NK92 and/or OTX015 administration) that we applied in our NSCLC xenograft model. The administration of OTX015 increased CD107a and IFN- γ expression in NK92, whereas impaired the expression of ICs, including PD1,CTLA4, and TIGIT.

Minor comments:

- Are there differences in phenotypes between CD56 bright and CD56 dim induced by BETi?

NKs were freshly isolated from tumor specimens and they exhibit a CD56 bright phenotype that is in line with their tissue localization. We did not appreciate the presence of an evident CD56 dim population in our samples.

- Some references didn't load in right- numbers 22 and 29, for example.

We carefully revised all references throughout the manuscript.

Reviewers' Comments:

Reviewer #1:

None

Reviewer #2:

Remarks to the Author:

The authors have performed additional experiments that have significantly improved the quality and translational impact of the manuscript.

Reviewer #3:

Remarks to the Author:

The resubmission of the manuscript by Reggiani has addressed all of the points raised by this reviewer, and in fact, all reviewers. Of note, the revised manuscript is strengthened in terms of the in vivo data and model systems employed and an expansion of the immune data. The clarity of the writing has also been improved.

Overall, I think the quality of the article meets the standards of Nature Communication, and I am please to promote its publication.

Reviewer #4:

Remarks to the Author:

The authors have adequately addressed my concerns. Also experimental data was presented for some key points

Overall the manuscript is much improved.

REVIEWERS' COMMENTS

Reviewer #2 (Remarks to the Author):

The authors have performed additional experiments that have significantly improved the quality and translational impact of the manuscript.

Reviewer #3 (Remarks to the Author):

The resubmission of the manuscript by Reggiani has addressed all of the points raised by this reviewer, and in fact, all reviewers. Of note, the revised manuscript is strengthened in terms of the in vivo data and model systems employed and an expansion of the immune data. The clarity of the writing has also been improved.

Overall, I think the quality of the article meets the standards of Nature Communication, and I am please to promote its publication.

Reviewer #4 (Remarks to the Author):

The authors have adequately addressed my concerns. Also experimental data was presented for some key points

Overall the manuscript is much improved.

Authors response:

We would like to thank all the Reviewers for positive consideration of our work.